# Plastic bending in a semiconducting coordination polymer crystal enabled by delamination

Lian-Cai An[1], Xiang Li[2], Zhi-Gang Li[1], Qite Li[3], Patrick J. Beldon [4], Fei-Fei Gao[1], Zi-Ying Li[1], Shengli Zhu [3], Lu Di[1], Sanchuan Zhao[1], Jian Zhu[1], Davide Comboni[5], Ilya Kupenko[2], Wei Li [1] ✉, U. Ramamurty [6,7] ✉ & Xian-He Bu [1] ✉

Coordination polymers (CPs) are a class of crystalline solids that are considered brittle, due to the dominance of directional coordination bonding, which limits their utility in flexible electronics and wearable devices. Hence, engineering plasticity into functional CPs is of great importance. Here, we report plastic bending of a semiconducting CP crystal, Cu-Trz (Trz = 1,2,3-triazolate), that originates from delamination facilitated by the discrete bonding interactions along different crystallographic directions in the lattice. The coexistence of strong coordination bonds and weak supramolecular interactions, together with the unique molecular packing, are the structural features that enable the mechanical flexibility and anisotropic response. The spatially resolved analysis of short-range molecular forces reveals that the strong coordination bonds, and the adaptive C−H···π and Cu···Cu interactions, synergistically lead to the delamination of the local structures and consequently the associated mechanical bending. The proposed delamination mechanism offers a versatile tool for designing the plasticity of CPs and other molecular crystals.

Coordination polymers (CPs), as a class of crystalline solids consisting of metal nodes and organic linkers, are at the forefront of materials science as they offer enormous application potential in electronics[1], magnetics[2], and optics[3]. However, their single crystals, constructed by directional coordination bonds, are intrinsically brittle, which severely limits their utility in applications such as flexible electronics and smart wearable devices[4,5]. It has been recently demonstrated that bendable organic crystals could be applied as deformable optical waveguides and circuits[6], flexible conductors[7], and ferroelectrics[8]. However, the few known bendable CPs have not been shown to exhibit any particular functionalities, in stark contrast to the vast number of existing CPs with attractive physical properties[9–13]. From the materials engineering perspective, therefore, making the CPs mechanically robust while preserving various functionalities is a major challenge.

A number of recent studies have shown that many organics[14–22] and a few CP crystals[4,9–13] and some coordination complexes[23–25] can be mechanically deformable, through both elastic and plastic bending. Atomistic-scale analysis via micro-focus X-ray diffraction and spectroscopic methods shows that a delicate balance of different kinds of intermolecular interactions is the main cause of the observed mechanical behavior[4,10,12,26–31]. For the elastic bending, two general structural features are found in these crystals. The first is the existence of abundant isotropic weak interactions, such as dispersive van der Waals interactions and weak hydrogen bonds[12,15,32]. The uniform and

[1]School of Materials Science and Engineering, Nankai University, 300350 Tianjin, China. [2]Institut für Mineralogie, University of Münster, Corrensstr. 24, 48149 Münster, Germany. [3]School of Materials Science and Engineering, Tianjin University, 300350 Tianjin, China. [4]Cortirio, NETPark, North East Technology Park, Sedgefield TS21 3FD, UK. [5]European Synchrotron Radiation Facility, 71 avenue des Martyrs, F-38000 Grenoble, France. [6]School of Mechanical and Aerospace Engineering, Nanyang Technological University, Singapore 639798, Republic of Singapore. [7]Institute of Materials Research Engineering, Agency for Science, Technology and Research, Singapore 138634, Republic of Singapore. ✉e-mail: wl276@nankai.edu.cn; uram@ntu.edu.sg; buxh@nankai.edu.cn

isotropic distribution of weak interactions in these crystals could provide an effective pathway to accommodate mechanical stress via small structural adaption in the elastic regime. In addition, these elastically bendable crystals normally have an interlocked crystal packing motif, which discourages long-range molecular sliding and lattice slippage, hence preventing plastic deformation and fracture over a significant range of stresses[20,21,33]. For the plastic bending systems, the interaction distributions (the strong and weak interactions) are usually found to be anisotropic and directional in the crystal structures, facilitating the crystal sliding along the directions of the weak interactions and resulting in the final plastic deformation[4,13,25]. These relatively weak interactions can be abundant in the lattice, often existing between adjacent layered molecular assemblies that are adhered via relatively strong directional bonds. Correspondingly, facile molecular sliding can be allowed in weak bonding directions while the structural scaffolding is maintained along the strong bonding orientation. Due to this special kind of interaction distribution, crystals can accommodate the imposed strain permanently through the slippage of the molecular layers past each other to maintain the crystal's integrity[4,13,34].

While there are some common mechanistic features of bending in molecular crystals and CPs, some distinct features were also reported[4,9–13]. In particular, the low structural dimensionality of CPs often plays a pivotal role in dictating their bendability[4,9]. In most known systems, the low dimensional coordination chains act as rigid skeletons and are assembled via abundant weak molecular interactions along the other two directions. Upon bending, the coordination skeletons remain significantly less affected since the mechanical stress is mainly resolved via the reorganization of the weak bonding forces.

In this context, it could be proposed that the introduction of combined low dimensional coordination connections and weak intermolecular interactions into an extended lattice could be a feasible strategy for enhancing the interaction anisotropy and achieving bendability in CPs. If this is possible, such a delicate leverage would greatly extend the landscape of flexible CP crystals, which would promote the rational design of such materials for practical applications. Towards this goal, we report herein a plastically bendable CP semiconductor, Cu-Trz (Trz = 1,2,3-triazole). Spatial analysis of the local structures shows that the plastic bending is enabled through the strong coordination bonds in combination with short-range adaptive supramolecular interactions. First-principles calculations confirm this deformation mechanism. On the basis of the experimental observations, a delamination-induced bending model is proposed to rationalize the plastic bending in Cu-Trz.

## Results

Cu-Trz crystallizes in the monoclinic $P2_1/c$ space group[35] and has a one-dimensional chained structure which is constructed by one Cu (I) and three triazolate ligands via coordination bonds (Fig. 1a). These chains stack closely via the Cu⋯Cu cuprophilic interactions[36] ($d = 3.05\,\text{Å}$) to further form a layered motif. Then, adjacent layers are joined together via the C−H⋯π interactions ($d = 2.81\,\text{Å}$) in a herringbone packing mode to form a 3D supramolecular structure. Interestingly, strong coordination bonds extend only along [101] while two kinds of relatively weak supramolecular interactions propagate along the two directions orthogonal to it. Taking account of the coexistence of both strong and weak bonds in the structure, slippage could occur along the chain and layer directions under local shear stress[37].

The plastic bending nature of Cu-Trz crystals was revealed by the three-point bend tests that were performed by loading on the different crystal faces[38]. Prior to the mechanical tests, face indexing was performed on the as-grown crystals which demonstrated that (010) and (10-1) planes are the most amenable ones for the bend tests (Fig. 1b and Supplementary Fig. 1a). When the (010) face was stressed, the crystals

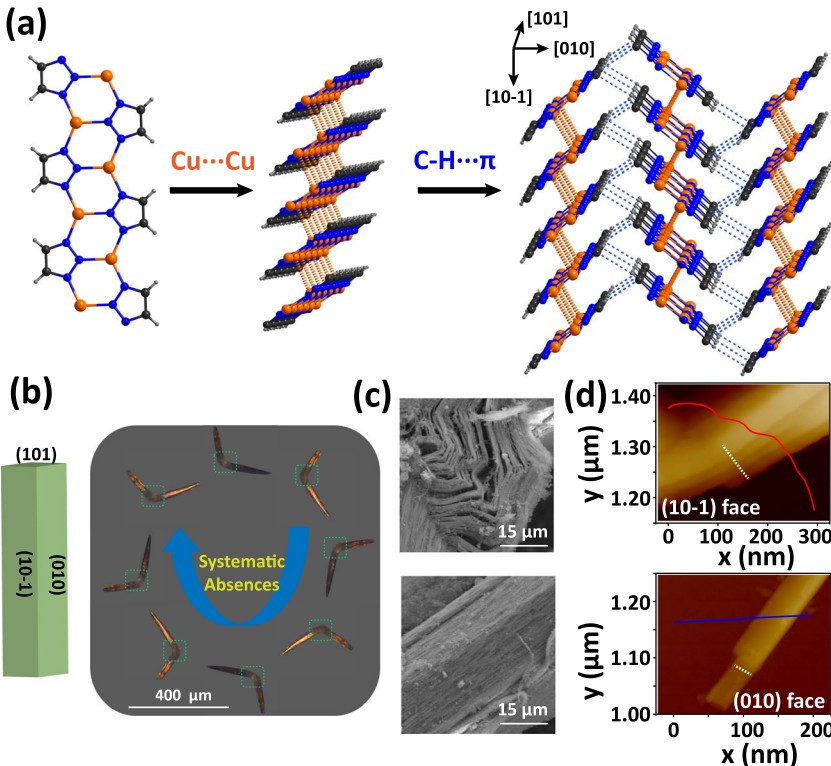

**Fig. 1 | Crystal structures and morphology of Cu-Trz. a** Crystal structures of Cu-Trz. The blue lines represent the C−H⋯π interactions and the orange dash lines represent the Cu⋯Cu interactions. Color scheme: C, black; N, blue; H, gray; Cu, orange. **b** The crystal face index and the polarizing microscopic images on the (10-1) face. **c** SEM images of (10-1) of the bent section (top) and the straight section (bottom). **d** Height profiles in (10-1) face (top) and (010) face (bottom) taken along the white dash lines of AFM micrographs (inset).

bend considerably, which remained after the removal of the stress, indicating its plastic nature. However, the external stress can readily fracture the crystals when the (10-1) face was stressed (Supplementary Fig. 1b).

Interestingly, the polarizing microscopy (Fig. 1b) showed that the undeformed straight sections of the crystals exhibit systematic light extinctions, a typical crystal characteristic in a full turn, while the bent sections do not. Such distinct polarizing changes indicate the loss of the long-range structural ordering and the disruption of the anisotropy in the bent sections. The lateral (10-1) face of the bending section was imaged by SEM (Fig. 1c), which reveals that there are two different straight sections and a bent section. Clearly, the bent section of the crystal becomes striped with obvious stratifications while the straight sections are integral and tight. The applied forces drive the layers to segregate which leads to delamination between the neighboring molecular layers. As shown in Fig. 1d and Supplementary Fig. 2, AFM imaging of the bent section on the (10-1) surface (top) illustrate that these originally densely stacked layers have delaminated, with the width of each layer being less than 100 nm. On the contrary, the topography of (010) surface is relatively flat without any fracture. Such delamination in the bent sections undoubtedly affects the deformed local structures[39,40]. Due to such different interaction strengths, significant anisotropy is also found in Young's modulus, linear compressibility, and shear modulus (Supplementary Figs 3–6, Supplementary Tables 1–2) via density functional theory (DFT) calculations[41–43]. Further discussion on these properties can be found in Supplementary Note 1.

To dynamically disentangle the bending mechanism, spatially resolved analysis of the local structures during the mechanical deformation was performed. First, laboratory micro-SCXRD was conducted to examine the crystallinity of the straight and bent sections precisely. As expected, the 2D diffraction frames (Supplementary Fig. 7) show sharp and discrete diffraction spots in the straight sections (top), indicating their good single crystallinity. When the X-ray is focused on the bent sections (bottom), the diffraction spots are significantly elongated and weakened, leading to the formation of diffraction rings, implying the deterioration of the long-range order in the deformed region of the bent crystal. Further, the reciprocal spaces (Supplementary Fig. 8) are reconstructed based on the diffraction frames and the bent sections similarly show stretching and weakening spots along with disrupted periodicity compared with those from the straight section.

Furthermore, we conducted synchrotron micro-X-ray diffraction measurements in the ID15B beamline ($\lambda = 0.41005$ Å), the European Synchrotron Radiation Facility with a beam size of $2 \times 4$ μm$^2$ to obtain spatially analysis of different regions (from the inner region, the middle region to the outer region). The same diffraction deterioration is also observed in the diffraction frames from the straight to the bend in Supplementary Fig. 9. As shown in Fig. 2, the profile shows a small but noticeable variation across the bend. The $a$- and $b$-axes extracted from the positions in both the inner and outer regions show slightly increased lengths. However, the $a$- and $b$- lengths are less affected in the middle region. From the crystal structure, it can be seen that the Cu⋯Cu and C–H⋯π interactions are approximately along the $a$- and $b$-axis, respectively, so the expansions of $a$- and $b$-axes in the inner and outer regions could be attributed to the elongation of Cu⋯Cu and C–H⋯π bonds. In terms of the middle region of the bend, the counterbalanced tensile and compressive strains lead to nearly unvaried $a$- and $b$- lengths. In addition, the bending strain on both the inner and outer regions elongates the Cu⋯Cu and C–H⋯π bonds which force the metal coordination chains to slip oppositely to result in decreased $c$-axis length from the middle to the inner/outer region. The slipping of the coordination chains also induces a slight overall increase of $\beta$ angle in both terminals of the bend. This result demonstrates that the bending stress could elongate the Cu⋯Cu and C–H⋯π bonds in both

the inner and outer regions so that the slippages could take place in both the interlayers and inter coordination chains. These two weak interactions can not only facilitate the microscale crystal sliding via adaptive changes, they can also alleviate and solve the bending strain via the macroscopic crystal delamination.

Micro-Raman spectroscopy was performed to obtain spatially resolved information of the adaptive molecular interactions. For this purpose, the outer, middle, and inner regions of the bent and the straight sections (insert in Fig. 3a) were probed. Firstly, we have performed DFT calculations which show that the low and high-wavenumber Raman bands can be attributed to the Cu–N out-of-plane bending and the stretching of the triazolate ligand (C–C, C–N and N–N stretching), respectively, as shown in Supplementary Fig. 10 and Supplementary Movie 1 in the Supplementary Data. Then the Raman results were analyzed by using the Lorentzian fitting. The peak positions and the full width at the half maximum (FWHM) of the straight, inner, middle and outer sections are shown in Supplementary Figs. 11–14 and Supplementary Tables 3–5. The peak positions of the four low frequency bands do not show any obvious changes in the straight and bent sections. This observation demonstrates that the Cu–N bonds did not experience any significant changes during bending so that the backbone of the 1D CP chains keep almost intact. However, the intensities of these low frequency bands decrease dramatically in the bend compared with those of the straight section. This implies the sliding of the 1D CP chains could be possibly due to the reorganization or even partial breakage of the interchain Cu⋯Cu and interlayer C–H⋯π interactions. In addition, FWHMs of the four low frequency bands in the bent section display an overall broadening which indicate that the Cu⋯Cu interactions adhering adjacent 1D CP chains into the layered motif could be weakened to lead to partial loss of the long-range structural order. The peak positions of the high frequency range also show negligible changes between the straight and bent sections. However, the respective intensities of the bent section increase substantially over those of the straight counterpart, indicating possible enhancement of the stretching of the triazolate ligand. This phenomenon could arise from the weakening of the C–H⋯π interactions under stress, which partly liberates the two H atoms and increases the vibrational freedom of their neighboring two C atoms in the triazolate ring, hence giving rise to enhanced C–C and C–N vibration intensities[44]. This is consistent with the partial loss of the long-range structural order explained above.

To further explore the changes in the bonding characteristics during such deformation, the crystal structures under different tensile strains (0, 2, and 8% tensile strains) were calculated using the LOBSTER code[45]. The electron localization function (ELF) was used to quantify the electronic localization states at different regions. As shown in Fig. 3c–e, the distance of Cu⋯Cu interactions ($d = 2.983$ Å) can adaptively change from $d = 3.034$ Å (under 2% tensile strain) to $d = 3.211$ Å (under 8% tensile strain). The crystal orbital Hamilton population (COHP) and the partial density of states (PDOS) of Cu⋯Cu interactions were calculated to reveal the bonding and anti-bonding states (Fig. 3f–h). The bonding states (-CHOP value is positive) of the tensile strained structure (around 1.65 eV below the Fermi level) are mostly attributed to the 3$d$ orbital electrons of the Cu atoms. The values of the integrated CHOP, iCHOP, of the 2 and 8% strained structures (−0.215 and −0.177 eV, respectively) are -6.1 and 22.7% more positive than that of the straight section (−0.229 eV), indicating the elongated and hence weakened Cu⋯Cu interactions[44]. Therefore, the strain that results from the applied stress on the crystal during bending can be efficiently resolved by the delamination.

The relative changes in the unit cell parameters in different crystallographic directions were investigated using high-pressure XRD (HP-XRD) for pressures up to 1.3 GPa[46,47]. As shown in Supplementary Fig. 15, both [10-1] with Cu⋯Cu interactions and [010] with C–H⋯π interactions decrease (−1.89% and −5.41%, respectively) upon

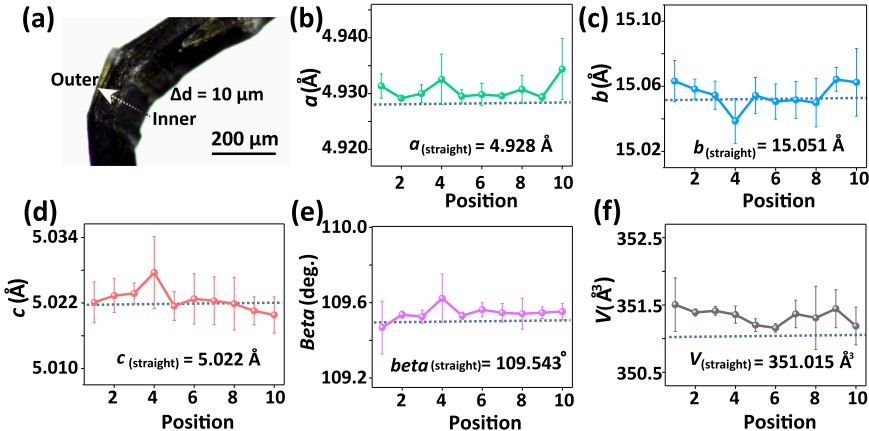

**Fig. 2 | Synchrotron X-ray diffraction analysis of the bend of a Cu-Trz crystal.** **a** Optical micrograph of a bent crystal. The insert scale bar represents 100 μm. **b**–**f** Cell parameters (*a*, *b*, *c*, *β*, *V*) extracted from the diffraction patterns at a 10 μm interval from the inner to the outer region of the bend. Note that, the error bars were obtained from three parallel experiments and the blue dashed lines represent the standard cell parameters of the straight section.

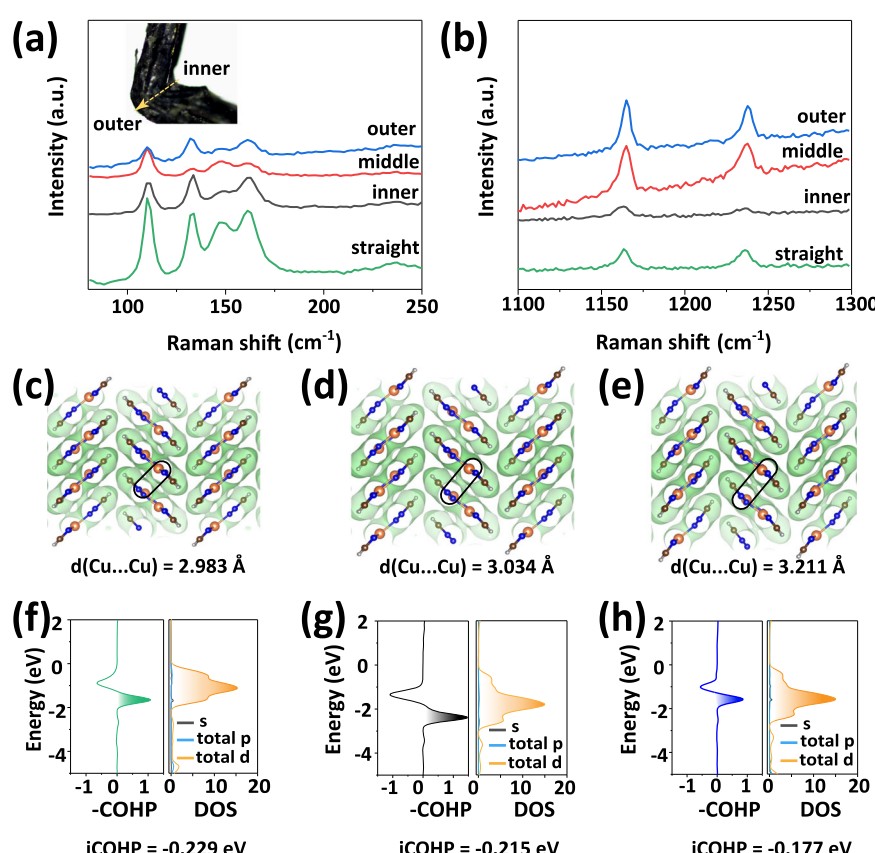

**Fig. 3 | Spatial micro-Raman shift analysis and theoretical calculations.** The low- (**a**) and high-wavenumber (**b**) Raman spectra were collected from the inner, middle, and outer regions of the bent and the straight section, respectively. The electron localization function (ELF) and the corresponding crystal orbital Hamilton population (COHP) and the partial density of states (PDOS) of Cu···Cu interactions of the undeformed (**c**, **f**), 2% (**d**, **g**) and 8% (**e**, **h**) tensile strained structures.

compression while the coordination chains along [101] increase marginally (1.28%). These results imply that both the C–H···π and Cu···Cu interactions are amenable to stimulation via the application of external pressure and adaptively change to maintain the crystal structure's integrity. Further discussion on the HP-XRD results and the negative linear compressibility (NLC) along [101] is provided in Supplementary Figs. 15–17, Supplementary Table 6 and Supplementary Note 2.

The different interactions that are present in the crystal structure respond differently to the hydrostatic pressure as well as the temperature. As shown in Supplementary Fig. 18 and Supplementary Tables 7–9, both the C–H···π and Cu···Cu interactions show relatively large changes in their interaction distance with the increasing temperature, compared with the marginal changes observed for Cu–N bonding. These results are consistent with the HP-XRD results. Interestingly, the crystals also exhibit negative thermal expansion (Supplementary Figs. 18–19) along [101]; see Supplementary Note 3 for details.

Nanoindentation experiments, conducted on the different facets of the Cu-Trz single crystals, further confirm their anisotropic

mechanical behavior (Supplementary Figs. 20–22). Based on the detailed discussion in Supplementary Note 4, the stress localization can produce delamination via breaking of the C–H···π bonds and the plastic deformation induced 'pile-up' occurs (Supplementary Figs. 20c, 22a). Such a scenario under localized stress during indentation is consistent with the crystal bending on the (010) face instead of brittle fracture.

Based on the spatially resolved analysis and previous reports[4,13], a model depicting the bending mechanism is proposed by adapting (Fig. 4). Before bending, the coordination chains are stacked into layers by the Cu···Cu interactions, and adjacent layers are further joined by the C–H···π interactions (Fig. 4a). When the crystal is bent on the (010) face, the adjacent coordination chains would slide over each other due to the relatively weak interchain Cu···Cu interactions, resulting in layer expansion along [101] in the bent section. Meanwhile, the slipping of the coordination chains causes concurrent weakening and even possible breakage and reconstruction of a large amount of C–H···π bonds, leading to significant sliding between adjacent layers. As both the Cu···Cu and C–H···π interactions are soft and weakly directional, the consequent facile chain and layer sliding enables significant malleability of the crystal under stress. Upon further bending, slipping is not enough to fully counterbalance the mechanical stress so that the breakage of the coordination chains occurs in the inner region of the (010) face. This drastic change leads to complete disruption of a large amount of C–H···π bonds and corresponding detached adjacent layers in the bent section. Finally, these detached layers bulge upward to result in delamination. Therefore, the rigid coordination bond and adaptive molecular interactions, with discrete nature along different directions in the crystal, are responsible for the delamination which can effectively relieve the local stress to give rise to the plastic bending.

The electronic band structures and the partial density of states (PDOS), obtained from the DFT calculations[48], are shown in Fig. 5a. It can be seen that Cu-Trz possesses an indirect band gap of 1.06 eV. As for the PDOS, the valence band maximum is mainly derived from the Cu-3$d$ and N-2$p$ states while the conduction band minimum is from the N-2$p$ states. To examine the change in the electrical conductivity ($\sigma$) of the Cu-Trz crystals under three different bent conditions (straight, 90° and 180° bent angles) experimentally, the two-contact probe method was utilized (as shown in Fig. 5c). The following equation was used to estimate $\sigma$:

$$\sigma = \frac{I}{V} \times \frac{L}{A} \tag{1}$$

where $I$ is the current, $V$ is the voltage, $L$ is the crystal length and $A$ is the cross-sectional area. $\sigma$ was measured. The current-voltage ($I$-$V$) plots (Fig. 5b) obtained on the three samples are similar, indicating their similar electrical conductivities ($\sigma_{straight} = 1.12$, $\sigma_{90°} = 1.19$, and $\sigma_{180°} = 1.07 \times 10^{-7}$ S/cm). This result suggests that the electric conductivity of Cu-Trz crystals does not get affected by the plastic deformation. The values of $\sigma$ are typical of semiconductors and are comparable with other metal triazolates M-(1,2,3-Trz) (M = Fe$^{2+}$, Co$^{2+}$, Cd$^{2+}$, Cu$^{2+}$, Mg$^{2+}$, Mn$^{2+}$, Zn$^{2+}$) (see Supplementary Table 10). Although the bent deformation breaks a significant amount of C–H···π interactions between the layers held by other two kinds of bonding forces, the crystals can still maintain a good $\sigma$ because the contribution of the C–H···π interactions is less to the band structure and the PDOS. This important characteristic of Cu-Trz might hold a great application potential in flexible and wearable devices.

## Discussion

In conclusion, we report a plastically bendable CP crystal which arises from the coexistence of different kinds of molecular bonding forces with distinct magnitudes of strengths. Our spatially dynamic analysis enabled by optical and electronic imaging techniques disclose that the strong coordination bonds and adaptive supramolecular interactions along different crystallographic directions lead to the delamination and consequent plastic bending of this single crystalline CP system. Our proposed mechanism of the plastic bending would stimulate further search of flexible CPs towards versatile potential applications in electronics and optoelectronics.

## Methods

### Synthesis and crystallization
All the reactants and reagents were commercially purchased and used directly without any further purifications. Cu-Trz crystals were synthesized according to literature[35] and the details were described as follows: a mixture of CuCl$_2$•2H$_2$O (0.102 g, 0.60 mmol) and 1,2,3-triazole (200 μL, 0.70 mmol) in 10 mL of distilled water was stirred for 10 min. After that, the mixture was sealed in a Teflon-lined reaction vessel and heated at 170 °C for 4 d. Then dark brown rod-like crystals were obtained.

### Three-point bending test
A carefully selected crystal was supported with forceps and a needle was approaching to the crystal then the crystal was bent at that point. The bending angle was ranging from 0 to 180°.

### General characterizations
The powder X-ray diffraction (PXRD) patterns were recorded with a Rigaku MiniFlex600 diffractometer (40 kV, 15 mA) equipped with a Cu-target tube and graphite monochromatic. The scanning electron microscopy (SEM) was performed on a JSM-7800F facility. The polarizing photos were carried out on a Nikon LV100NPOL microscope.

### Single-crystal X-ray diffraction (SC-XRD)
A good single crystal was selected to perform the SC-XRD on a Rigaku XtaLAB MM007 CCD diffractometer (Mo Kα $\lambda = 0.71073$ Å). Data collection and structural refinement were performed using the Rigaku CrysAlisPro and Olex2 software package[49,50]. The variable temperature SC-XRD experiments were performed in the temperature range from 100 to 400 K at a 25 K interval. The thermal expansion coefficients were calculated using the PASCal software[51] based on the cell parameters at different temperatures.

### Laboratory micro-focus XRD
The lab micro-focus X-ray diffraction was measured using a Rigaku XtaLAB MM007 CCD diffractometer (Cu Kα, $\lambda = 1.54056$ Å) with a focal spot size of ~70 × 70 μm$^2$. A carefully bent Cu-Trz crystal, with a length of ~1 mm and a width of ~100 μm, was measured with an incident X-ray beam normal to the bending plane. The data collections were set with an exposure time of 0.5 s, the theta range of 6–73° and the cell parameters were refined using the Olex2 software package.

### Synchrotron micro-focus XRD
X-ray diffraction measurements with a micro-ray beam of 2 × 4 μm$^2$ were performed in the ID15B beamline ($\lambda = 0.41005$ Å), the European Synchrotron Radiation Facility. The X-ray beam was incident normal to the bending plane, and the ω range and the exposure time were set as ±10°, 2.5 s, respectively. The diffraction images were collected using the *Eiger2 9 M CdTe* detector. The measured crystal was moved by 10 μm from the inner regions to the outer regions and parallel experiments were repeated three times. The lattice parameters were refined with measured images using the UnitCell software[52].

### Atomic force microscopy (AFM)
The surface topographies and the corresponding height profiles of different surfaces were collected in the ScanAsyst Mode with a BRUKER RTESPA-300 tip. The samples were prepared by diluting the Cu-Trz fine powders in ethanol to a concentration of 0.1 mg/mL, and then

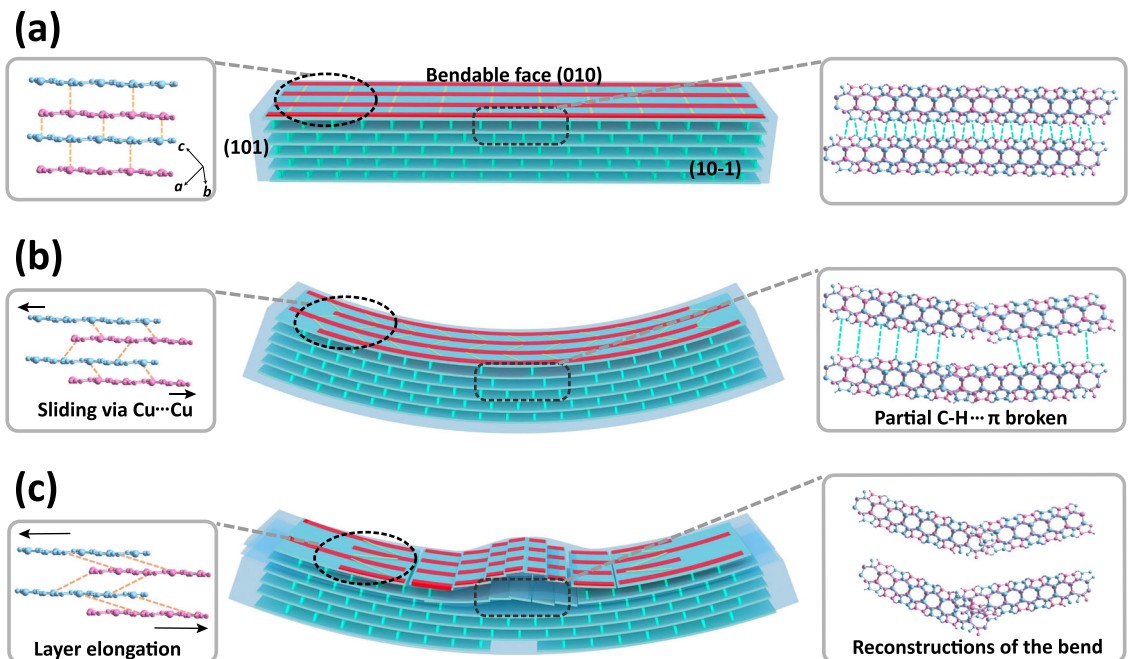

**Fig. 4 | The proposed delamination mechanism of the Cu-Trz crystal during the plastic bending.** The schematic diagrams and crystal structures before (**a**), during (**b**), and after bending (**c**). Note that the red, yellow and green lines represent the coordination chains, Cu···Cu interactions, and the C−H···π interactions between the layers, respectively.

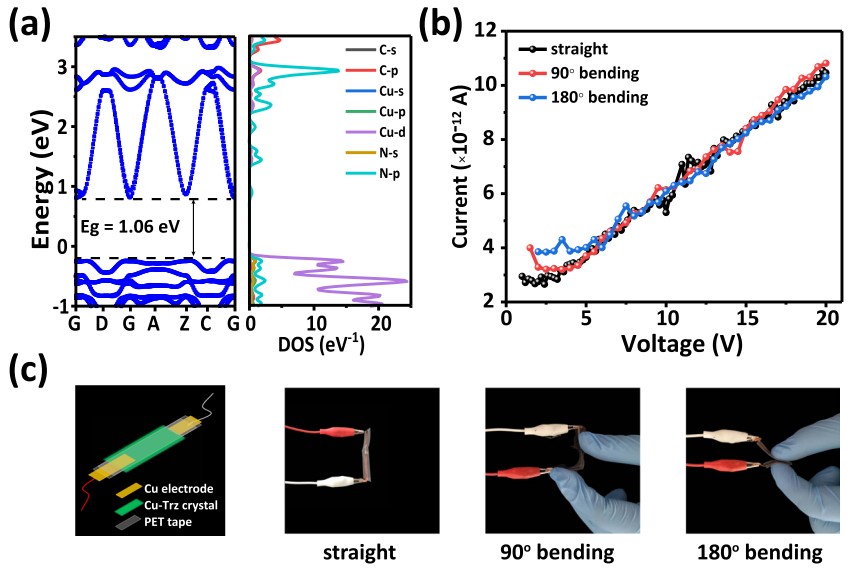

**Fig. 5 | The band structure and conductivity of crystal. a** Calculated electronic band structures and densities of states for Cu-Trz. **b** The current-voltage (*I-V*) curves of different bending crystals. **c** The experiments scheme of the two-contact probe method and the different states (straight, 90° bending and 180° bending) of bending crystals.

the milky suspension in the vial was centrifuged at a rate of 3000 r/min for 3 min. Keeping the tube still for a few minutes, the top suspension containing microcrystals was dipped on a preheated silicon wafer ($0.8 \times 0.8$ cm²). After complete evaporation of the ethanol on the substrate, the as-prepared microcrystals were scanned at a rate of 1 Hz. The images were taken with a minimum resolution of $256 \times 256$ pixel[2].

### High-pressure powder X-ray diffraction (HP-PXRD)

Firstly, the hydrostatic pressure was generated by the systematic diamond anvil cells (DACs) with a culet diameter of 400 μm and the silicone oil was served as the pressure-transmitting medium. Then the well-ground powder samples and ruby spheres were placed in a hole of ~190 μm diameter in a pre-indented stainless-steel gasket[53] with a thickness of ~40 μm. The generated pressures were quantitatively measured by measuring the fluorescence shift of the ruby as a function of pressure. The in situ HP-PXRD was performed at the Beijing Synchrotron Radiation Facility (BSRF) with a wavelength of 0.61992 Å. The measured pressure range for Cu-Trz was 0–2.591 GPa, the diffraction patterns were collected by Pilatus3 2 M detector and further processed by the FIT2D software package. The cell parameters were obtained by Le Bail refinement using the Total Pattern Solution (TOPAS) software[42]. Based on these cell

parameters, the bulk modulus and the linear compressibilities were calculated using the PASCal software.

## Nanoindentation

Firstly, the face indexation was conducted on single crystals using a Rigaku XtaLAB diffractometer to determine its natural faces at room temperature. Then the carefully selected single crystals were cold mounted in epoxy resin and polished well before the nanoindentation test[54,55]. The nanoindentation experiments were performed at ambient temperature with a three-sided pyramidal Cube-Corner nanoindenter (Triboindenter of Hysitron, TI-Premier, USA; tip radius = 50 nm). The loading/unloading rates were maintained at 0.2 mN/s and kept the holding time for 10 s at peak load. Nine indents were performed on the different faces to determine the Young's moduli ($E$) and hardnesses ($H$). The average values of $E$ and $H$ were obtained from the $P$-$h$ curves using the standard Oliver-Pharr Method[56].

## Micro-focus Raman

Micro-focus Raman spectra of different regions (straight, inner regions, middle regions, and outer regions) were collected on a Horiba LabRAM HR Evolution micro-Raman spectrometer with a 600 lines/mm grating and 532 nm laser with the beam size of ~2 μm and accumulated twice with exposing 10 s at each time. In details, a ×50 long working distance Olympus microscope objective was used to focus the laser beam and all the Raman tests are conducted in the room temperature.

## Electronical conductivity

Individual well-selected rod-like crystals were fixed onto a piece of tape, and two thin copper film electrodes were put on the top of each crystal. To achieve good contact between the copper electrodes and the crystal, a drop of sliver glue was applied on the interface. Finally, another piece of tape was put to seal the device. The electronical conductivity was measured using a Keithley 4200A-SCS parameter analyzer at room temperature.

## Density functional theory (DFT) calculations

All of our DFT calculations were performed as implemented in the Vienna Ab initio Simulation Package (VASP) with a plane-wave cutoff of 520 eV[57] and the projector augmented wave (PAW)[58] pseudopotentials were employed. LDA was used as the exchange-correlation energy functional to optimize the atomic positions and lattice parameters until the total energy converged to within $10^{-8}$ eV and the residual force on each atom was less than 0.01 eV/Å. DFT-D3 Grimme correlation was adopted to account for dispersion effects. The Brillouin zone integration was used with the $3 \times 2 \times 3$ $k$-point mesh sampling scheme of the Monkhorst-Pack for geometry optimization and the calculations of elastic constants. The elastic constants were calculated by using Stress-Strain Method[59] and the number of strains was set to 7. The Gaussian smearing method was used for self-consistent fields and band structure calculations, and the tetrahedron method with Blochl corrections was used for PDOS calculations. The calculations of biaxial strain were achieved by multiplying the $a$ and $c$ unit vectors by 1.02 (1.08) and fixing them, while the $b$ unit vector and ionic positions were able to relax (user-modified VASP version).

The crystal orbital Hamilton population (COHP) and integrated COHP (iCOHP) were calculated using the LOBSTER code[60] to analyze the electronic and bonding state in Cu-Trz. Noting that, the negative COHP (i.e., -COHP) and the positive COHP were corresponding to the bonding interactions and anti-bonding interactions, respectively. Furthermore, the projection parameters were set as C 2p 2 s; Cu-3d 4 s; H 1s; N-2p 2 s. As a result, the charge spilling coefficient was 1.23% indicating the reliable calculation.

The Raman off-resonant activity was obtained by calculating the polarizability for the vibrational modes using VASP code as the backend[61].

## Data availability

The other relevant data are available from the authors upon request. Source data are provided with this paper.

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

## Acknowledgements

We acknowledge the financial support from the National Natural Sci-ence Foundation of China (Nos. 21975132, 22035003, and 21991143) and the Fundamental Research Funds for the Central Universities (No. 63196006). The high-pressure powder X-ray diffraction measurements were performed at beamline 4W2, Beijing Synchrotron Radiation Facility (BSRF). We also acknowledge the European Synchrotron Radiation Facility for the provision of synchrotron radiation facilities of the micro-X-ray diffraction measurements and Susanne Müller from University of Münster of for experimental support.

## Author contributions

W.L., U.R., and X.-H.B. conceived the idea and designed the experi-ments. L.-C.A. carried out the synthesis of Cu-Trz, performed three-point bending, VT-SCXRD, laboratory micro-focus SC-XRD, HP-PXRD mea-surements. P.J.B. contributed this raw idea. X.L., D.C., and I.K. helped to performed the micro-X-ray synchrotron diffraction and X.L. also per-formed the Raman test. Z.-G.L. carried out first-principles DFT

calculations. L.-C.A. and Z.-G.L. analyzed the theoretical results. Q.L. and S.Z. (Shengli Zhu) performed the nanoindentation tests. L.D. performed the SEM experiments, S.Z. (Sanchuan Zhao) and J.Z. conducted the electronical conductivity of the bent crystals. L.C.A. wrote the manuscript and F.-F.G. and Z.-Y.L. help to revise pictures and formats. All authors discussed and commented on the manuscript.

## Competing interests

The authors declare no competing interests.
