## [Peer Review File · Nature Communications]

Plastic bending in a semiconducting coordination polymer crystal enabled by delaminationREVIEWER COMMENTS

Reviewer #1 (Remarks to the Author):

Thank you for the opportunity to review the manuscript by An et al. 'Adaptive Supramolecular Interactions Induced Delamination Enables Plastic Bending in a Semiconductive Coordination Polymer Crystal'. The manuscript reports a very interesting mechanically flexible 1D coordination polymer based on a Cu triazole backbone that exhibits a remarkable degree of plasticity. The authors have conducted a thorough analysis of the bending mechanism using myriad analytical tools, including some insights from *ab initio* simulations. The authors demonstrate that plastic bending results from delamination of layers within the CP crystal, which itself stems significant anisotropy of the intermolecular interactions within the structure. Additionally, the authors have demonstrated that their flexible CP is electronically conductive, and remains as such in its bent state, thereby opening the door to flexible electronic devices based on CP materials. While I am certain this contribution will be very welcomed by the community, I regret that I cannot yet recommend its publication in the current form until the authors have addressed some concerns.

Some Major Concerns:

1. The authors have claimed to have measured micro-focus XRD on their crystals. However, there is no indication in the paper or ESI how these measurements were performed. In particular, there is no indication as to the size of the beam. It is therefore not possible to assess the validity of the claims being made.
2. The authors have provided two 2D diffraction images of the straight and bent crystals (Figure 2). However, the authors have not provided any indication as to the relative 'cut' through reciprocal space being seen. It is therefore not possible to reliably compare 'changes', although there is a clear indication of mosaicity in the bent crystals that is absent from the straight crystals. Moreover, the integrated 1D plots seem to show different sets of reflections. What, for example, are the indexed reflections in the bent crystal? What has happened to reflections at ca. 9, 12, 15.5, 16.5, 19.5 degrees? On this same note, the authors claim that reflections have shifted to lower angles during bending. However if the integrated regions are identical in Fig 2b, it seems that a number of reflections may have shifted to higher angles (lower d spacing). This discussion needs significant clarification.
3. The microfocus Raman spectra are indeed quite interesting, although not entirely convincing. It seems inconsistent that all regions of the crystal exhibit identical Raman shifts $< 300\text{ cm}^{-1}$ except the 'inner' arc. The optical image of the single crystal seems to suggest a fragment of the crystal has 'detached from the crystal under the black dot – could this be the origin of the apparent vibrational band? How reproducible is this effect, if a second crystal is analysed? On this note, the authors have also drawn somewhat haphazard curved lines in Figure 3c for frequencies in the region $1100\text{--}1300\text{ cm}^{-1}$. Any change in the frequencies would be much more convincing with a proper fit to the bands, and explanation of why the 'straight' crystal is consistently softer than the bent regions, even for regions that should be 'elongated' such as the outer arc. On this note, I am also not convinced that one should see changes in 'hard' intramolecular vibrational modes on account of deformation in the lattice. Can the authors comment further on this? Is there any simulation evidence to support the nature of the low frequency + high frequency bands? The authors also claim that the bands are 'broader' in the bent region, but this is not convincing by looking solely at Figure 3.
4. The authors interpret the low frequency Raman spectra, particularly with bands $< 200\text{ cm}^{-1}$, based on Goreshnik et al. (Ref 32). However, this paper does not seem to have any indication of Cu-N coordinative bonds, which the authors ascribe bands to in the present paper. I apologise if I have missed this in Ref 32, but perhaps the authors should revisit their interpretation. It seems unusual to have Cu-N coordination bonds at such low frequencies (see e.g. Dudley et al 1973 J. Inorg. Chem.) which places these modes around 450 cm^{-1} . Presumably the bands $< 200\text{ cm}^{-1}$ are external lattice modes? I also do not see convincing evidence for the red and blue shift of the 242 cm^{-1} band (line 156) in Figure 3b (presumably not Figure 3c as stated on line 157).
5. The crystal packing in the crystal structure (CIF OFUZUL) indicate that the CP chains run along the long axis of the crystal (i.e. along [101]). This means that contraction and expansion of the layers can be only accomplished by compression / elongation of the CP chains themselves (i.e. the network that

runs along the crystal). The Cu...Cu interactions occur perpendicular to the crystal bending axis, and therefore do not appear to be able to account for the 'contraction of the inner layers' as suggested by the authors in line 197-198. If I do not misunderstand the figure (it is also not clear from the caption), it seems to me then that the mechanism proposed in Figure 4 cannot be correct as the red bars (presumably the CP chains?) should run in the plan of the page (left to right) with the non-covalent interactions running into the page. An alternative model based on 'rigid' CP backbones has been proposed to solve this problem by Bhattacharya et al (2020), developed further by Dakovic et al. (2021, 2022). Perhaps the authors could consider something in this direction, unless a more suitable alternative can be developed.

Some Minor Concerns:

1. The referencing in the introduction has a few issues. Notably, reference 8 (which the authors state 'reported to exhibit tunable magnetic properties; Line 40) only theorises this effect based on simulation, but does not experimentally demonstrate this. Reference 11 is not a coordination polymer, but rather a metal-organic complex. Ref 11 should be removed from the list of 'flexible CPs' being discussed (Line 42). In reference to microfocus X-ray diffraction and spectroscopy (line 49), the authors should include citation of works related to CPs that have already implemented these techniques, including Ref 4 (Bhattacharya et al.), Ref 12 (Liu et al.) and recent work by Dakovic and colleagues (e.g. doi.org/10.1021/acs.chemmater.1c00539)
2. In the abstract (Line 17) the authors state that CPs are 'mechanically undeformable'. The authors should be careful here, as there is significant work done in high dimensional CPs (e.g. MOFs) which exhibit remarkable mechanical deformability. See work e.g. by FX Coudert, S Moggach, T Bennet, etc.
3. On line 64 of the introduction the authors discuss strategies to manipulate flexibility in molecular crystals. Perhaps it is their intention to name CPs here?
4. On line 82 of the introduction the authors state that the herringbone packing is demonstrative of plasticity. In fact, this packing motif is most common of elastically flexible crystals, at least in the field of molecular crystals. Perhaps it is worth a note on this from the authors in the introduction.
5. The authors show photos of the bent crystals under polarized microscope (Fig 1b). Although I understand the intention, as this is a common technique to explore 'single crystallinity' of crystals, the authors have not accounted for the fact that the crystal is already geometrically distributed in the polarized beam (i.e. the straight portions are at ca 45° with the bent at 90°). I would therefore expect these regions of the crystal to appear differently under a polarized microscope, regardless of defects. Although the figures are very nice, this does not appear to be particularly convincing evidence for structural distortion in the region of the crystal.
6. Regarding SEM images, these seem to be much more significant proof of the delamination than optical microscopy, but do not seem to be referenced or discussed in the main text. Moreover, there is no indication in the caption as to which faces of the crystal are being measured, and it is not immediately obvious the connection between 1d and 1b (although with some deeper thought this can be understood). Perhaps a comment in the caption can help guide the reader to reaching this connection more easily.
7. Perhaps for the non-specialist further discussion of the AFM results could be helpful. For example why do the authors claim a connection between the flatness of the surface and the degree of delamination? Similarly, what does the AFM of the surface look like for an entirely unbent (control) crystal – is there evidence that stress from the bend does not propagate? Moreover, the authors have indicated that the AFM profiles are measured on the (top) (010) and (bottom) (10-1) faces. However, these are the bending faces, and one would expect to see any delamination on the orthogonal faces. Please can the authors verify which faces were in fact measured?
8. The authors have performed high pressure XRPD analysis. What is the relevance, particularly in light of Liu et al. 2021 (Nat Commun), which suggests that HP behaviour and uniaxial compression during bending are disparate?
9. The authors have conducted an interesting electronic structure analysis using LOBSTER and the COHP method. It would be helpful if the authors could provide more information on their methodology in the ESI, namely the projection parameters (set of orbitals for projection) and the spilling coefficients that are used to assess the quality of the projection.
10. The authors could please provide more detailed information on their simulations – which

pseudopotentials were used (presumably PAW potentials if LOBSTER simulations were performed?). Similarly, to allow for assessment of the reliability of elastic constants, it would be helpful for the authors to provide a comparison of their optimized crystal geometry as compared with the low temperature experimental data. I note also that the authors should be careful about their claims based on a PBE band gap (here 1.37 eV), as GGA functionals are very well known to grossly underestimate fundamental band gaps. That said, this seems to have no real implications for the science presented here.

11. The authors interpret the lack in change of conductivity based on CH... π interactions. However, I am not surprised to find a lack of change in the electrical conductivity primarily because the only 'pathway' in CPs is through covalent networks – i.e. the CP chains themselves. Electrical conductivity through NCIs is very poor. The lack of change might therefore instead indicate that no real disruption occurs to the CP networks themselves? This could go a long way to understanding whether chains are themselves perturbed during bending in 1D CPs, or whether something more like the model proposed by Bhattacharya (2020) is more in the right direction.

Reviewer #2 (Remarks to the Author):

The paper explores the interesting behaviour of a plastically flexible semi-conducting coordination polymer. The science presented is currently in a hotly contested cutting-edge area of chemistry. In this respect it is suitable for publication in Nature Communications. The authors have also performed some very nice experimental work suitable for publication.

That said, there are some very significant issues with the manuscript that would require major revision before being considered acceptable for this or indeed any other reputable journal.

1) As I noted above, this area is currently at the cutting edge of research. Not surprisingly perhaps there is some considerable disagreement in the literature about the nature of mechanisms of flexibility and suitable methods for confirming them. Most of the mechanisms of flexibility presented in the literature are based almost entirely on speculation made from considering the crystal structures of natural (unbent) crystals. Methods for unequivocally determining mechanisms using SCXRD have now been demonstrated for elastic and plastic crystals. See for example Worthy, A. et al. Atomic resolution of structural changes in elastic crystals of copper(II) acetylacetonate. *Nature Chem.* 10, 65-69, doi:10.1038/nchem.2848 and Bhandary, S. et al. The mechanism of bending in a plastically flexible crystal. *Chem. Commun.* 56, 12841-12844, doi:10.1039/D0CC05904H (2020).

Both papers are referenced in the manuscript but Worthy et al. is cited as an example of a coordination polymer with attractive properties. This paper is actually the first to report an experimentally determined mechanism of elastic flexibility and should be referenced accordingly. There is nothing in this paper about coordination polymers. Worthy et al. should be referenced with the other references at the end of the sentence beginning line 47 " Atomistic-scale analysis via micro-focus X-ray diffraction...". It would also be appropriate to reference the recent review in ChemSocRev on this topic Thompson, A. J. et al. Elastically flexible molecular crystals. *Chem. Soc. Rev.*, doi:10.1039/D1CS00469G (2021). While the latter is about elastic crystals, there is a lot of information therein about micro-focus XRD for determination of mechanisms applicable to both plastic and elastic crystals.

2) It is also important to delineate which examples are of plastic deformation and which are elastic. Line 39 reads "Meanwhile a flexible metal complex was reported to exhibit tunable magnetic properties (ref 8)." This example is of an elastic crystal. So the question is how is the manipulation of properties in this example relevant to the example in the MS which undergoes plastic deformation?

The above points 1 and 2 are just a couple of examples but the entire introduction of the manuscript needs to be re-written and carefully referenced to properly reflect the current state of the literature.

3) The authors of the present manuscript have made an attempt to determine the mechanism of

plastic flexibility in the current system using both micro focus SCXRD and Raman spectroscopy however in this case neither approach gives detailed and unequivocal structural confirmation of the mechanism because the XRD data could not be indexed and the Raman data alone does not provide enough information to do determine structures explicitly. This is OK – but the authors need to make it very clear that the proposed mechanism for plastic deformation is a hypothesis and not proven. In the concluding paragraph of the paper and in the abstract and in the title, the authors overstate the certainty of their findings.

4) The following sentences beginning on line 52 are very difficult to follow. “Specifically, the abundant former could allow molecules to rotate in a reversible manner under stress while the small number of latter are able to redistribute the applied mechanical strain via limited changes in the elastic regime. In the plastically bendable crystals, the irreversible deformation is mainly induced by the adaptive but unrecoverable changes of dispersive van der Waals and weakly directional C–H... π interactions (ref 22).” Is there a clearer way to say all this? Note that the mechanism of plastic flexibility is almost always associated with dislocation along slip planes.

5) Line 90 – The authors indicate that the crystals remain bent after stress and that this “indicates plastic (or permanent) nature”. A relatively minor point but plastic and permanent are not interchangeable. Not all permanent deformations are plastic (e.g. if a crystal breaks or delaminates etc that process is not plastic but it is still permanent).

6) Figure 1. (c) The picture of crystals bent in to letters to advertise a University has been done to death and adds no additional scientific value. What does bending the crystals into these letters show that bending them into any other letters or numbers or any other random shape does not show?

7) Line 113-114 shear modulus repeated.

8) Line 125-6 The inability to index, solve and refine the X-ray data means all subsequent proposals for mechanisms are purely hypotheses. As stated earlier this needs to be fixed in the title, abstract, conclusions etc to make this clear throughout the MS.

9) There is some nice powder data presented on page 7. Can the authors comment on how much the shift in the peaks from a straight section of crystal to a bent section could be caused by delamination? Could delamination at the microscopic level result in peak broadening? What about the shift in peaks?

10) Blue light is approx 25000 cm^{-1} while red light is approx 15000 cm^{-1} . With this in mind almost all of the discussion about Raman spectra, peak positions and in particular peak shifts on page 9 of the MS is wrong. The concept of blue and red shifting can only make sense for spectral bands between 15000 and 25000 cm^{-1} . In the manuscript the peaks discussed are between 200 and 1300 cm^{-1} . For example, it makes no sense to say a shift from 1168 to 1173 cm^{-1} is “blue shifted” as such a peak is also shifting towards the red region of the spectrum and is therefore “red shifted” too. It would be more informative to talk about shifts to higher/lower energy or frequency.

11) The discussion of the results of the high pressure crystallography perhaps needs some clarification. The authors note that the Cu- π and π - π distances change markedly with pressure while the Cu-N bond remains virtually unchanged. I am not sure what the significance of this is in terms of plastic bending. This result is consistent with everything we know about crystallography. Contacts in which the interactions are described by broad and shallow potential energy wells e.g. π - π interactions are much more compressible than covalent and coordinate bonds that are described by deep narrow potential energy wells? Why is this relevant to the plastic deformation? There are no circumstances in which you would expect the Cu-N bond to significantly expand or contract enough to facilitate plastic flexibility.

12) Sentence ending 198 – see other comments about being clear that the “adaptable reconstruction” is a hypothesis consistent with the data – not proven unequivocally.

13) Lines 201-204 I found this sentence confusing. Doesn't the process of delamination imply loss of crystal integrity? It is hard for me to see how delamination can take place while at the same time the crystal integrity (i.e. long range order) is retained?

14) Line 212-213 The authors state that bending requires elongation of the inner layers and shortening of the outer layers. I think this is around the wrong way? In any case this elongation and shortening only applies to elastic bending. In plastic crystals there is no requirement for the inside or outside arcs to lengthen or shorten.

Supporting Information

Line 117 of the MS: The authors discuss results of micro-SCRD experiments in the MS but there are

no details provided about the experimental set-up used in the supporting information. This information is vital. What was the cross-sectional diameter of the X-Ray beam? What was the relationship of the beam to the crystal? How many data sets collected and across what angle? The authors should provide sufficient detail so that the experiment could in principle be repeated by others. Also – without this information it is not possible to properly referee the results presented. For example a highly focussed beam say 10 microns orthogonal to the bent edge will give a very different result to a wider beam and or if it is oriented in an alternate direction with respect to the crystal.

As I said above this is nice work in a cutting-edge field. If the authors were to provide a new version addressing all the concerns above then I think the MS would be suitable for publication in Nature Comms.

Reviewer #3 (Remarks to the Author):

I have read this manuscript with interest. The paper contains much detailed and interesting data. It definitely deserves publication in a good journal. I would have no doubts to recommend it for publication in *Crystal Growth and Design*, *CrystEngComm*, *IUCrJ*, and many other top-journals. I do not see any reasons why this paper could not be published also in *Nature Communications*: even though this is not the first example of a highly plastic crystal, that has been studied by a plethora of advanced experimental and computational techniques, such studies are still rare, very challenging, and always have a high impact.

My advise of improvement would be to add the discussion of a few references that seem to be quite relevant for this work:

Rath, B. B., & Vittal, J. J. (2021). Mechanical bending and modulation of photoactuation properties in a one-dimensional Pb (II) coordination polymer. *Chemistry of Materials*, 33(12), 4621-4627.

Thomas, S. P., Shi, M. W., Koutsantonis, G. A., Jayatilaka, D., Edwards, A. J., & Spackman, M. A. (2017). The Elusive Structural Origin of Plastic Bending in Dimethyl Sulfone Crystals with Quasi-isotropic Crystal Packing. *Angewandte Chemie International Edition*, 56(29), 8468-8472.

Pejov, L., Panda, M. K., Moriwaki, T., & Naumov, P. (2017). Probing structural perturbation in a bent molecular crystal with synchrotron infrared microspectroscopy and periodic density functional theory calculations. *Journal of the American Chemical Society*, 139(6), 2318-2328.

Arhipov, S. G., Losev, E. A., Nguyen, T. T., Rychkov, D. A., & Boldyreva, E. V. (2019). A large anisotropic plasticity of L-leucinium hydrogen maleate preserved at cryogenic temperatures. *Acta Crystallographica Section B: Structural Science, Crystal Engineering and Materials*, 75(2), 143-151.

Feiler, T., Michalchuk, A. A., Schröder, V., List-Kratochvil, E., Emmerling, F., & Bhattacharya, B. (2021). Elastic Flexibility in an Optically Active Naphthalidenimine-Based Single Crystal. *Crystals*, 11(11), 1397.

Paliwoda, D., Wawrzyniak, P., & Katrusiak, A. (2014). Unwinding Au+... Au+ bonded filaments in ligand-supported gold (I) polymer under pressure. *The Journal of Physical Chemistry Letters*, 5(13), 2182-2188.

Das, S., Saha, S., Sahu, M., Mondal, A., & Reddy, C. M. (2021). Temperature-Reliant Dynamic Properties and Elasto-Plastic to Plastic Crystal (Rotator) Phase Transition in a Metal Oxyacid Salt. *Angewandte Chemie International Edition*, e202115359

Morritt, G. H., Michaels, H., & Freitag, M. (2022). Coordination polymers for emerging molecular devices. *Chemical Physics Reviews*, 3(1), 011306.

RESPONSES TO REVIEWERS' COMMENTS

Manuscript ID: ADDMA-D-22-01992

We sincerely thank the reviewers for their valuable comments and constructive suggestions on our manuscript. We have made point-to-point responses (in blue) and highlighted the changes (in yellow) in the revised manuscript.

Reviewer #1 (Remarks to the Author):

Thank you for the opportunity to review the manuscript by An et al. 'Adaptive Supramolecular Interactions Induced Delamination Enables Plastic Bending in a Semiconductive Coordination Polymer Crystal'. The manuscript reports a very interesting mechanically flexible 1D coordination polymer based on a Cu triazole backbone that exhibits a remarkable degree of plasticity. The authors have conducted a thorough analysis of the bending mechanism using myriad analytical tools, including some insights from ab initio simulations. The authors demonstrate that plastic bending results from delamination of layers within the CP crystal, which itself stems significant anisotropy of the intermolecular interactions within the structure. Additionally, the authors have demonstrated that their flexible CP is electronically conductive, and remains as such in its bent state, thereby opening the door to flexible electronic devices based on CP materials. While I am certain this contribution will be very welcomed by the community, I regret that I cannot yet recommend its publication in the current form until the authors have addressed some concerns.

Response:

We thank the reviewer for these enthusiastic comments on the novelty, significance, and quality of our work. We have carefully revised our manuscript to address the reviewer's concerns as detailed below.

Some Major Concerns:

1. The authors have claimed to have measured micro-focus XRD on their crystals.

However, there is no indication in the paper or ESI how these measurements were performed. In particular, there is no indication as to the size of the beam. It is therefore not possible to assess the validity of the claims being made.

Response:

We are sorry for not including the experimental details of the micro-focus XRD. We have added detailed description of the micro-focus SC-XRD experiments using a lab facility on page 2 in the Supporting Information as the following:

“The lab micro-focus X-ray diffraction was measured using a Rigaku XtaLAB MM007 CCD diffractometer (Cu $K\alpha$, $\lambda = 1.54056 \text{ \AA}$) with a focal spot size of $\sim 70 \times 70 \mu\text{m}^2$. A carefully bent **Cu-Trz** crystal, with a length of $\sim 1 \text{ mm}$ and a width of $\sim 100 \mu\text{m}$, was measured with an incident X-ray beam normal to the bending plane. The data collections were set with an exposure time of 0.5 s, the theta range of $6\text{--}73^\circ$ and the cell parameters were refined using the Olex2 software package.”

Supplementary Fig. 9 | Synchrotron diffraction patterns of different regions of a bent Cu-Trz crystal. (a) The obtained diffraction pattern of the straight section. (b-d) Diffraction patterns of the inner, middle, and outer regions of the bent section.

Fig. 2. | Synchrotron X-ray diffraction analysis of the bend of a Cu-Trz crystal. (a) Optical micrograph of a bent crystal. The insert scale bar represents $100 \mu\text{m}$. (b-f) Cell parameters (a , b , c , β , V) extracted from the diffraction patterns at a $10 \mu\text{m}$ interval from the inner to the outer region of the bend. Note that, the error bars were obtained from three parallel experiments and the blue dashed lines represent the standard cell parameters of the straight section.

While our lab micro-focus X-ray results can give some evidence to distinguish the straight and bent sections of the **Cu-Trz** crystals, we felt high precision synchrotron radiation data could provide more compelling evidence after receiving the reviewers' comments. Accordingly, we have conducted synchrotron micro-focus X-ray diffraction measurements in the ID15B beam line ($\lambda = 0.41005 \text{ \AA}$) of the European Synchrotron Radiation Facility. The incident beam with a cross-section of $2 \times 4 \mu\text{m}^2$ was normal to the bending plane of **Cu-Trz** crystals, and the diffraction images were collected at 10 equidistant positions from the inner, middle to the outer regions of the bend with a $10 \mu\text{m}$ interval. The diffraction patterns from different regions of the bend were shown in the new **Supplementary Figs. 9a-d**. The results can be summarized as the following: (1) The diffraction patterns in new **Supplementary Figs. 9a-d** show a gradually deteriorated diffraction in the sequence of the inner \rightarrow middle \rightarrow outer regions though the long range structural order is still partially preserved. Compared with the sharp

reflections observed from the straight section, the diffraction images obtained from the bent regions show a strong diffuse feature. (2) From the series of diffraction images collected from synchrotron radiation, the spatial profiles of the cell parameters were extracted and are shown in **Figs. 2b-f**, which demonstrate a small but noticeable variation across the bend. The *a*- and *b*-axes extracted from positions on both the inner and outer regions show a slight increase. However, the *a*- and *b*- lengths are less affected for the middle region. From the crystal structure, it can be seen that the Cu···Cu and C–H··· π interactions are approximately along the *a*- and *b*- axes, respectively, so the expansions of *a*- and *b*- axes in the inner and outer regions could be attributed to the elongation of Cu···Cu and C–H··· π bonds. For the middle region of the bend, the presence of a neutral axis results in nearly unvaried *a*- and *b*- lengths. In addition, the bending strain on both the inner and outer regions elongates the Cu···Cu and C–H··· π bonds which forces the adjacent metal coordination chains to slip past each other that, in turn, results in the decreased *c*-axis length from the middle to the inner/outer region. The slip of the coordination chains also induces a slight overall increase of β angle in both terminals of the bend.

2. The authors have provided two 2D diffraction images of the straight and bent crystals (Figure 2). However, the authors have not provided any indication as to the relative ‘cut’ through reciprocal space being seen. It is therefore not possible to reliably compare ‘changes’, although there is a clear indication of mosaicity in the bent crystals that is absent from the straight crystals. Moreover, the integrated 1D plots seem to show different sets of reflections. What, for example, are the indexed reflections in the bent crystal? What has happened to reflections at ca. 9, 12, 15.5, 16.5, 19.5 degrees? On this same note, the authors claim that reflections have shifted to lower angles during bending. However, if the integrated regions are identical in Fig 2b, it seems that a number of reflections may have shifted to higher angles (lower *d* spacing). This discussion needs significant clarification.

Response:

Based on the X-ray diffraction frames collected from the bent and straight sections, we have constructed the reciprocal spaces to investigate the changes between them. As shown in the figure below, the straight section shows strong and discrete diffraction spots that correspond to all the $(hk0)$, $(h0l)$ and $(0kl)$ planes, indicating good single crystallinity therein. However, the diffraction spots of the $(hk0)$ planes from the bent section exhibit significant stretching and weakening along with disrupted periodicity compared with those from the straight section. In the $(h0l)$ planes of the bent section, many new satellite diffraction spots appear which are away from the original positions in the reciprocal lattice. For instance, the $(0-100)$ reflection in the $(0kl)$ plane of the bent section shows an obvious smearing compared with that of the straight section, which indicates that the uniform inter-layer spacing in the original state diverges to nonuniform distances in the bend. These deteriorated diffraction images unambiguously suggest that the bent section partially loses the original long-range ordering.

Supplementary Fig. 8 | The diffraction patterns of straight (top) and bent (bottom) sections collected from a lab facility. The reciprocal spaces are reconstructed based on the X-ray diffraction frames in the $(hk0)$, $(h0l)$, and $(0kl)$ planes. The insert scale bar: 0.1 mm.

In terms of the PXRD patterns, the 1D plots of the straight and bent section are extracted from frames measured by a lab facility with a beam size of $\sim 70 \times 70 \mu\text{m}^2$. This beam size is relatively large (compared with that of a synchrotron facility), which inevitably covers mixed inner, middle and outer regions of the bent section during data collection. Therefore, the Bragg peak positioning and intensity integrating of any individual region is unable to be precisely quantified. In other words, it is not accurate to conclude that the Bragg peaks of the bent section shift towards lower angles as the reviewer pointed. To avoid any confusion, we have removed the discussion about the 1D PXRD plots in the revised text and corresponding data in the Supplementary Data. Instead, we have conducted the synchrotron micro-focus X-ray diffraction measurements and focused our discussion on the corresponding results as mentioned above.

3. The microfocus Raman spectra are indeed quite interesting, although not entirely convincing. It seems inconsistent that all regions of the crystal exhibit identical Raman shifts $< 300 \text{ cm}^{-1}$ except the ‘inner’ arc. The optical image of the single crystal seems to suggest a fragment of the crystal has ‘detached from the crystal under the black dot – could this be the origin of the apparent vibrational band? How reproducible is this effect, if a second crystal is analysed? On this note, the authors have also drawn somewhat haphazard curved lines in Figure 3c for frequencies in the region 1100-1300 cm^{-1} . Any change in the frequencies would be much more convincing with a proper fit to the bands, and explanation of why the ‘straight’ crystal is consistently softer than the bent regions, even for regions that should be ‘elongated’ such as the outer arc. On this note, I am also not convinced that one should see changes in ‘hard’ intramolecular vibrational modes on account of deformation in the lattice. Can the authors comment further on this? Is there any simulation evidence to support the nature of the low frequency + high frequency bands? The authors also claim that the bands are ‘broader’ in the bent region, but this is not convincing by looking solely at Figure 3.

Response:

We thank the reviewer for these critical comments. To exclude the effect of the ‘fragment detached from the crystal’ and confirm the reproducibility of the Raman results, we have re-collected data from 3 different crystals as shown below. It can be seen that the Raman spectra from all three crystals are consistent, which have peaks at about 107–110 (peak #1), 131–133 (peak #2), 146–148 (peak #3) and 161–162 (peak #4) cm^{-1} in the low frequency range, and 1162–1166 (peak #5) and 1235–1239 (peak #6) cm^{-1} in the high frequency range, respectively.

Supplementary Fig. 11 | The collected Raman spectra from 3 different Cu-Trz crystals. The optical micrographs and the low-wavenumber and high-wavenumber Raman spectra for crystal #1 (a-c), crystal #2 (d-f), and crystal #3 (g-i).

As suggested by the reviewer, we have done DFT calculations which disclose that the low and high wavenumber Raman bands are respectively attributed to the Cu–N out-

of-plane bending and the stretching of the triazolate ligand (C–C, C–N and N–N stretching) as shown in **Supplementary Fig. 10** below and **Movie S1** in the Supplementary Data. In addition, our calculated Raman spectra match well with the experimental results as seen in **Supplementary Fig. 10** below.

With the newly collected reproducible data, the Raman results have been carefully re-analyzed by using the Lorentzian fitting, and the peak positions and widths (the full width at the half maximum, FWHM) of the straight, inner, middle and outer sections are shown in **Supplementary Figs. 12 to 14** and **Tables 3 to 5**.

Supplementary Fig. 10 | (a) The Raman vibrational modes calculated using DFT; (b) Experimental (from crystal #3) and calculated Raman spectra of **Cu-Trz**. The inset shows the normalized experimental and calculated Raman bands at the low frequency.

Supplementary Fig. 12 | The Lorentzian fitting of the Raman bands collected from crystal #1.

Supplementary Table 3 | The peak positions and FWHMs obtained by the Lorentzian fitting of Raman data collected from crystal #1.

Peak position (cm⁻¹)	Straight	inner	middle	outer
Peak #1	110.8	110.8	110.2	110.0
Peak #2	133.0	133.3	132.8	132.6
Peak #3	147.8	147.4	147.5	147.2
Peak #4	161.7	161.8	161.7	161.2
Peak #5	1163.5	1162.3	1164.4	1164.8
Peak #6	1235.7	1235.3	1236.5	1237.7

FWHM (cm⁻¹)	Straight	inner	middle	outer
Peak #1	3.5	4.4	5.1	5.2
Peak #2	5.0	5.5	4.3	5.7
Peak #3	12.2	11.1	11.9	11.0
Peak #4	11.7	14.4	13.7	10.2
Peak #5	6.6	9.6	7.2	5.7
Peak #6	10.1	9.2	8.0	6.1

Supplementary Fig. 13 | The Lorentzian fitting of the Raman bands collected from crystal #2.

Supplementary Table S4 | The peak positions and FWHMs obtained by the Lorentzian fitting of Raman data collected from crystal #2.

Peak position (cm⁻¹)	Straight	inner	middle	outer
Peak #1	107.6	108.3	107.8	108.8
Peak #2	131.4	132.0	131.7	132.0
Peak #3	155.6	155.9	151.8	150.5
Peak #4	1163.6	1166.2	1166.4	1164.3
Peak #5	1234.7	1239.1	1236.6	1236.9

FWHM (cm⁻¹)	Straight	inner	middle	outer
Peak #1	7.2	8.3	10.5	6.8
Peak #2	12.2	13.0	10.5	5.6
Peak #3	20.2	21.5	23.5	27.0
Peak #4	10.7	5.5	9.7	9.8
Peak #5	10.8	5.5	10.0	10.7

Supplementary Fig. 14 | The Lorentzian fitting of the Raman bands collected from crystal #3.

Supplementary Table S5 | The peak positions and FWHMs obtained by the Lorentzian fitting of Raman data collected from crystal #3.

Peak position (cm⁻¹)	Straight	inner	middle	outer
Peak #1	110.3	110.0	110.8	110.6
Peak #2	133.0	132.8	133.2	133.3
Peak #3	147.4	146.0	146.6	146.9
Peak #4	161.6	160.5	161.9	162.0
Peak #5	1165.8	1165.9	1165.9	1166.2
Peak #6	1238.7	1238.9	1239.0	1239.4

FWHM (cm⁻¹)	Straight	inner	middle	outer
Peak #1	5.2	5.1	5.0	6.7
Peak #2	5.5	6.8	6.9	7.5
Peak #3	12.6	12.4	9.7	13.8
Peak #4	12.1	19.4	16.1	12.3
Peak #5	5.3	5.3	5.2	5.1
Peak #6	5.5	5.3	5.2	5.0

For the four low frequency bands, their peak positions do not show any obvious changes in the straight and bent sections which demonstrate the Cu–N bonds do not experience significant changes during the bending so that the backbone of the 1D CP chains keep almost intact upon the three-point bending. However, the intensities of these four Raman bands decrease dramatically in the bend compared with the straight section, which implies the sliding of the 1D CP chains could be possibly due to the reorganization or even partial breakage of the inter-chain Cu···Cu and inter-layer C–H··· π interactions. In addition, the FWHMs of bands #1-4 in the bent section display an overall broadening which indicate that the Cu···Cu interactions adhering adjacent 1D CP chains into the layered motif could be weakened to lead to partial loss of the long-range structural order.

In terms of peaks #5 and #6, the peak positions also show negligible changes between the straight and bent sections. However, the respective intensities of the bent section increase substantially over those of the straight counterpart, indicating possible enhancement of the stretching of the triazolate ligand. This phenomenon could arise

from the weakening of the C–H $\cdots\pi$ interactions under stress, which partly liberates the two H atoms and increases the vibrational freedom of their neighboring two C atoms in the triazolate ring, hence giving rise to enhanced C–C and C–N vibration intensities. This is consistent with the partial loss of the long-range structural order explained above.

4. The authors interpret the low frequency Raman spectra, particularly with bands < 200 cm⁻¹, based on Goreshnik et al. (Ref 32). However, this paper does not seem to have any indication of Cu-N coordinative bonds, which the authors ascribe bands to in the present paper. I apologies if I have missed this in Ref 32, but perhaps the authors should revisit their interpretation. It seems unusual to have Cu-N coordination bonds at such low frequencies (see e.g. Dudley et al 1973 J. Inorg. Chem.) which places these modes around 450 cm⁻¹. Presumably the bands < 200 cm⁻¹ are external lattice modes? I also do not see convincing evidence for the red and blue shift of the 242 cm⁻¹ band (line 156) in Figure 3b (presumably not Figure 3c as stated on line 157).

Response:

We thank the reviewer for pointing out our inappropriate citation of Reference 32 about the low frequency Raman modes. As the reviewer suggested, we have reconsidered the assignment of the above Raman peaks. According to literature and our theoretical calculations, the Raman bands in the region of 100–200 cm⁻¹ and 250–300 cm⁻¹ are assigned to the Cu–N and Cu \cdots Cu out-of-plane bending, respectively. To confirm the shift of the 242 cm⁻¹ band in the bent section, we have conducted additional Raman spectroscopy experiments for 3 individual crystals and re-analyzed the data as mentioned above. From the figure below, it can be seen that the 242 cm⁻¹ band is rather weak which was only able to be detected on the straight section of the bent crystals. Although this differs from the conclusion drawn in our previous manuscript, it does support that the original Cu \cdots Cu contacts in a perfect crystal were disrupted in the bent

section.

Fig. R1 The low-wavenumber Raman spectra are collected from crystal #1 (a), crystal #2 (b), and crystal #3 (c). The peaks marked with dashed lines are attributed from Cu...Cu interactions.

5. The crystal packing in the crystal structure (CIF OFUZUL) indicate that the CP chains run along the long axis of the crystal (i.e. along [101]). This means that contraction and expansion of the layers can be only accomplished by compression / elongation of the CP chains themselves (i.e. the network that runs along the crystal). The Cu...Cu interactions occur perpendicular to the crystal bending axis, and therefore do not appear to be able to account for the ‘contraction of the inner layers’ as suggested by the authors in line 197-198. If I do not misunderstand the figure (it is also not clear from the caption), it seems to me then that the mechanism proposed in Figure 4 cannot be correct as the red bars (presumably the CP chains?) should run in the plan of the page (left to right) with the non-covalent interactions running into the page. An alternative model based on ‘rigid’ CP backbones has been proposed to solve this problem by Bhattacharya et al (2020), developed further by Dakovic et al. (2021, 2022). Perhaps the authors could consider something in this direction, unless a more suitable alternative can be developed.

Response:

We agree with the reviewer that the $\text{Cu}\cdots\text{Cu}$ interactions are unable to account for the ‘contraction of the inner layers’ since the CP chains run along the $[101]$ direction. We are also very sorry for our incorrect mechanism described in **Fig. 4** (please see below) due to the misplacing of CP chains in the wrong direction. We agree with the reviewer that the model initially proposed by Bhattacharya et al (2020) would be a better starting point to understand the bending mechanism. Based on Bhattacharya’s model, we have re-organized Fig. 4 to explain the plastic bending facilitated by delamination in the **Cu-Trz** crystal.

Fig. 4 | The proposed delamination mechanism of the Cu-Trz crystal during the plastic bending. The schematic diagrams and crystal structures before (a), during (b) and after bending (c). Note that the red, yellow and green lines represent the coordination chains, $\text{Cu}\cdots\text{Cu}$ interactions, and the $\text{C-H}\cdots\pi$ interactions between the layers, respectively.

Some Minor Concerns:

1. The referencing in the introduction has a few issues. Notably, reference 8 (which the authors state ‘reported to exhibit tunable magnetic properties; Line 40) only theorises this effect based on simulation, but does not experimentally demonstrate this.

Reference 11 is not a coordination polymer, but rather a metal-organic complex. Ref 11 should be removed from the list of ‘flexible CPs’ being discussed (Line 42). In reference to microfocus X-ray diffraction and spectroscopy (line 49), the authors should include citation of works related to CPs that have already implemented these techniques, including Ref 4 (Bhattacharya et al.), Ref 12 (Liu et al.) and recent work by Dakovic and colleagues (e.g. doi.org/10.1021/acs.chemmater.1c00539)

Response:

We agree with the reviewer and have carefully reorganized the introduction as well as the references therein. The old Ref. 8 has been deleted for the inappropriate citation and new Refs. 10 and 13 have been included (please see the following and the revised text).

Ref 10: PISAČIĆ, M. et al. Elucidating the origins of a range of diverse flexible responses in crystalline coordination polymers. *Chem. Mater.* **33**, 3660–3668 (2021).

Ref 13: PISAČIĆ, M. et al. Two-dimensional anisotropic flexibility of mechanically responsive crystalline cadmium(II) coordination polymers. *Chem. Mater.* **34**, 2439–2448 (2022).

2. In the abstract (Line 17) the authors state that CPs are ‘mechanically undeformable’. The authors should be careful here, as there is significant work done in high dimensional CPs (e.g. MOFs) which exhibit remarkable mechanical deformability. See work e.g. by FX Coudert, S Moggach, T Bennet, etc.

Response: We thank the reviewer for this constructive comment and have changed “mechanically undeformable” with “brittle”.

3. On line 64 of the introduction the authors discuss strategies to manipulate flexibility in molecular crystals. Perhaps it is their intention to name CPs here?

Response: We thank the reviewer for pointing out this confusion and have changed “molecular crystals” with “CPs”.

4. On line 82 of the introduction the authors state that the herringbone packing is

demonstrative of plasticity. In fact, this packing motif is most common of elastically flexible crystals, at least in the field of molecular crystals. Perhaps it is worth a note on this from the authors in the introduction.

Response: We thank the reviewer for this constructive comment. We add a note about herringbone packing in elastic bending crystal: “In addition, these elastically bendable crystals normally have an interlocked crystal packing motif, which discourages long range molecular sliding and lattice slippage, hence preventing plastic deformation and fracture over a significant range of stresses.” Although there have been some papers (*Angew. Chem. Int. Ed.* 2020, **59**, 5557–5561, *Angew. Chem. Int. Ed.* 2017, **56**, 8468–8472) reporting the plastically bendable crystals with the herringbone packing motifs, it might not be a common characteristic for all plastic bending cases. Therefore, we have decided to remove the previous description about the herringbone packing of **Cu-Trz**.

5. The authors show photos of the bent crystals under polarized microscope (Fig 1b). Although I understand the intention, as this is a common technique to explore ‘single crystallinity’ of crystals, the authors have not accounted for the fact that the crystal is already geometrically distributed in the polarized beam (i.e. the straight portions are at ca 45° with the bent at 90°). I would therefore expect these regions of the crystal to appear differently under a polarized microscope, regardless of defects. Although the figures are very nice, this does not appear to be particularly convincing evidence for structural distortion in the region of the crystal.

Response:

As the reviewer mentioned, the polarized microscope is a standard technique to distinguish whether a material is singly refracting (optically isotropic) or doubly refracting (optically anisotropic). By rotating the stage by a full turn (360°), the polarization changes of both the bent and straight sections were recorded. As shown in the revised Fig. 1b (please see below), the straight sections show bright-to-dark changes with the rotation, which is a characteristic of single crystallinity. However, the bent section shows insignificant change with the rotation, indicating the disruption of the

anisotropy.

Fig. 1b. The polarizing microscopic images in a full turn.

6. Regarding SEM images, these seem to be much more significant proof of the delamination than optical microscopy, but do not seem to be referenced or discussed in the main text. Moreover, there is no indication in the caption as to which faces of the crystal are being measured, and it is not immediately obvious the connection between 1d and 1b (although with some deeper thought this can be understood). Perhaps a comment in the caption can help guide the reader to reaching this connection more easily.

Response:

We agree with the reviewer that the SEM images are more robust evidence of the delamination, so we have included some more SEM images (please see below) from parallel sets of experiments to further confirm the bending mechanism. As seen in **Supplementary Fig. 1** (please see below), the straight and bent sections show tightly packing and cleaving feature, respectively. The applied forces drive the layers segregating which leads to a delamination between the neighboring layers as evidenced from the repeatable SEM images. In addition, we have included specific discussions

about the SEM data, along with proper reference on paragraph 1, page 6, in the revise manuscript. Furthermore, we are sorry for not indicating the bending face in the caption of Fig. 1d. We have mentioned that the bending face is (010) and included additional text to connect Figs. 1b and 1d in page 5, in the revise manuscript.

Supplementary Fig. 1 | (c) SEM images collected from 4 individually bent crystals.

7. Perhaps for the non-specialist further discussion of the AFM results could be helpful. For example, why do the authors claim a connection between the flatness of the surface and the degree of delamination? Similarly, what does the AFM of the surface look like for an entirely unbent (control) crystal – is there evidence that stress from the bend does not propagate? Moreover, the authors have indicated that the AFM profiles are measured on the (top) (010) and (bottom (10-1) faces. However, these are the bending faces, and one would expect to see any delamination on the orthogonal faces. Please can the authors verify which faces were in fact measured?

Response:

We agree with the reviewer that further discussion of the AFM data is necessary. Firstly, we are very sorry for mis-labelling the two AFM profiles in old **Fig. 1e** (and its caption), in which the top and bottom faces should have been (10-1) and (010), respectively. Our original purpose is to demonstrate the different tomographies of the delaminated (10-1)

face in the bend and the nearly unaffected (flat) (010) face in the straight section which are consistent with the SEM images shown in **Fig. 1c** (the (10-1) and (010) are largely delaminated and corrugated). Secondly, we have further collected AFM images on an entirely bent crystal as suggested by the reviewer (please see figure below). During transferring and fixing this bent crystal on the substrate, it became twisted with (10-1) on the top right straight section, (010) on the left bottom straight section, and twisted (10-1) in the bent section, respectively. It can be seen that the stress is accumulated in the bent to result in apparent layer cleavage and does not propagate to the straight sections. Thirdly, we have emphasized that the bending face is (010) in the revision, and the delamination indeed occurs on the orthogonal (10-1) face.

Supplementary Fig. 2 | The AFM image of the bent crystal.

8. The authors have performed high pressure XRPD analysis. What is the relevance, particularly in light of Liu et al. 2021 (Nat Commun), which suggests that HP behaviour and uniaxial compression during bending are disparate?

Response:

As different kinds of chemical bonds respond disparately upon hydrostatic compression, HP-XRD has been considered as a useful means to investigate the bonding anisotropy in crystalline materials (*Natl. Sci. Rev.*, **2020**, 7, 149; *ACS Cent. Sci.*, **2016**, 2, 201). From our HP-XRD results between 0–1.3 GPa, the [10-1] (approximately along

Cu···Cu bonding) and [010] (approximately along C–H··· π bonding) directions respectively shrink by 1.89 and 5.41%, indicating the bonding strength: Cu···Cu > C–H··· π interactions. Considering a uniaxial stress applied on the **Cu-Trz** crystal, the slippage would tend to occur more easily along the weaker C–H··· π bonding direction rather than the stronger Cu···Cu bonding direction. This is fully consistent with the occurrence of delamination of the (010) face viewed along the [10-1] direction. In this regard, our HP-XRD results can further support the strong bonding anisotropy in **Cu-Trz** which is the main cause of the plastic bending as proved by other spectroscopic and microscopic measurements.

The crystal reported by Liu et al. 2021 (Nat Commun) is a tetragonal 1D coordination polymer (space group *P-4b2*), in which adjacent coordination chains are linked via same C–H···Cl interactions along the [110] and [-1-10] directions. Due to its isostructural crystal packing, it is not a surprise that this crystal shows a quite isotropic behavior under hydrostatic pressure. However, **Cu-Trz** has a highly anisotropic bonding and packing mode, which enables it to respond very anisotropically to pressure. Both the plastically bendable nature under 3-point bending and anisotropic response under pressure of **Cu-Trz** crystal are attributed to its intrinsic bonding anisotropy.

9. The authors have conducted an interested electronic structure analysis using LOBSTER and the COHP method. It would be helpful if the authors could provide more information on their methodology in the ESI, namely the projection parameters (set of orbitals for projection) and the spilling coefficients that are used to assess the quality of the projection.

Response:

We have added the detailed information of the LOBSTER and COHP calculations in the ESI as the following: “The COHP and integrated COHP (iCOHP) were calculated using the LOBSTER code. The negative COHP (i.e. -COHP) and the positive COHP (+COHP) were corresponding to the bonding and anti-bonding interactions,

respectively. The projection parameters were set as C 2p 2s; Cu 3d 4s; H 1s; N 2p 2s. As a result, the charge spilling coefficient was 1.23%, indicating the reliability of our calculations.”

10. The authors could please provide more detailed information on their simulations— which pseudopotentials were used (presumably PAW potentials if LOBSTER simulations were performed?). Similarly, to allow for assessment of the reliability of elastic constants, it would be helpful for the authors to provide a comparison of their optimized crystal geometry as compared with the low temperature experimental data. I note also that the authors should be careful about their claims based on a PBE band gap (here 1.37 eV), as GGA functionals are very well known to grossly underestimate fundamental band gaps. That said, this seems to have no real implications for the science presented here.

Response:

We have added the detailed information about our calculations in the ESI. The PAW pseudopotential was employed in the GGA frame. As the reviewer suggested, we have compared the optimized crystal geometry with the experimental crystal structure at 100 K in **Supplementary Table 9**. The optimized cell parameters ($a = 4.894 \text{ \AA}$, $b = 14.931 \text{ \AA}$, $c = 4.978 \text{ \AA}$, $\beta = 108.814^\circ$) match well with the experimental data ($a = 4.864 \text{ \AA}$, $b = 14.581 \text{ \AA}$, $c = 5.010 \text{ \AA}$, $\beta = 108.803^\circ$), indicating the reliability of our elastic constant calculations.

Fig. R2. The bandgap and the densities of states (DOS) using the PBE functionals (a), the HSE06 functionals (b), the LDA functionals (c) and the LDA + U functionals (d).

In addition, we agree with the reviewer that the GGA functionals underestimate the bandgaps and often cause inaccuracy. To evaluate the bandgap of **Cu-Trz** in a more accurate way, we have recalculated the bandgap and band structure using different functionals as shown in **Fig. R2** above. The calculated bandgaps differ greatly from 1.06 eV (LDA bandgap) to 2.77 eV (HSE06 bandgap). To validate our theoretical calculations, we have conducted UV-Vis-NIR spectroscopy to experimentally investigate the bandgap. As shown in **Fig. R3** below, the UV-Vis-NIR absorption spectra demonstrate the onset peak is located approximately at 1000 nm and the bandgap obtained by the Tauc plot method is about 1.24 eV. This experimental value more approximates the bandgap of 1.06 eV calculated using the LDA functionals (which also underestimate the bandgaps), so we have finally decided to use the LDA bandgap.

Fig. R3. UV-Vis-NIR absorption spectroscopy (a) and the corresponding Tauc plot (b).

11. The authors interpret the lack in change of conductivity based on CH... π interactions. However, I am not surprised to find a lack of change in the electrical conductivity primarily because the only ‘pathway’ in CPs is through covalent networks – i.e. the CP chains themselves. Electrical conductivity through NCIs is very poor. The lack of change might therefore instead indicate that no real disruption occurs to the CP networks themselves? This could go a long way to understanding whether chains are themselves perturbed during bending in 1D CPs, or whether something more like the model proposed by Bhattacharya (2020) is more in the right direction.

Response:

We agree with the reviewer that the electrical conductivity primarily runs along the CP chains and apologize for our inappropriate interpretation of the lack in change of conductivity based on C–H... π interactions. According to the bending model shown in the revised Fig. 4, the CP chains would be fractured during bending, which is supported by our microscopic experiments. Although some individual CP chains would be snapped upon bending, the fractured two parts are still closely contacted with each other which could partially keep the electrical conductivity. As the intrinsic conductivity of the intact **Cu-Trz** crystals is already quite low, it is possible that the bending induced change in conductivity is not significant enough to be measurable.

Reviewer #2 (Remarks to the Author):

The paper explores the interesting behavior of a plastically flexible semi-conducting coordination polymer. The science presented is currently in a hotly contested cutting-edge area of chemistry. In this respect it is suitable for publication in Nature Communications. The authors have also performed some very nice experimental work suitable for publication. That said, there are some very significant issues with the manuscript that would require major revision before being considered acceptable for this or indeed any other reputable journal.

Response:

We thank the reviewer for these enthusiastic comments on the significance and quality of our work. We have carefully revised the manuscript and ESI to address the questions raised by the reviewer.

1) As I noted above, this area is currently at the cutting edge of research. Not surprisingly perhaps there is some considerable disagreement in the literature about the nature of mechanisms of flexibility and suitable methods for confirming them. Most of the mechanisms of flexibility presented in the literature are based almost entirely on speculation made from considering the crystal structures of natural (unbent) crystals. Methods for unequivocally determining mechanisms using SCXRD have now been demonstrated for elastic and plastic crystals. See for example Worthy, A. et al. Atomic resolution of structural changes in elastic crystals of copper(II) acetylacetonate. Nature Chem. 10, 65-69, doi:10.1038/nchem.2848 and Bhandary, S. et al. The mechanism of bending in a plastically flexible crystal. Chem. Commun. 56, 12841-12844, doi:10.1039/D0CC05904H (2020). Both papers are referenced in the manuscript but Worthy et al. is cited as an example of a coordination polymer with attractive properties. This paper is actually the first to report an experimentally determined mechanism of elastic flexibility and should be referenced accordingly. There is nothing in this paper

about coordination polymers. Worthy et al. should be referenced with the other references at the end of the sentence beginning line 47 “Atomistic-scale analysis via micro-focus X-ray diffraction...”. It would also be appropriate to reference the recent review in ChemSocRev on this topic Thompson, A. J. et al. Elastically flexible molecular crystals. *Chem. Soc. Rev.*, doi:10.1039/D1CS00469G (2021). While the latter is about elastic crystals, there is a lot of information therein about micro-focus XRD for determination of mechanisms applicable to both plastic and elastic crystals.

Response:

We are sorry for citing the Nature Chem (2017, 10, 65) paper in an inappropriate place, and have moved it to the references regarding “Atomistic-scale analysis via micro-focus X-ray diffraction” as suggested by the reviewer. In addition, we have also cited the recent review paper (*Chem. Soc. Rev.*, 2021, 50, 11725) as new Ref. 15 in the revised manuscript.

Ref. 15, Thompson, A. et al. Elastically flexible molecular crystals. *Chem. Soc. Rev.*, **50**, 11725–11740 (2021).

2) It is also important to delineate which examples are of plastic deformation and which are elastic. Line 39 reads “Meanwhile a flexible metal complex was reported to exhibit tunable magnetic properties (ref 8).” This example is of an elastic crystal. So the question is how is the manipulation of properties in this example relevant to the example in the MS which undergoes plastic deformation? The above points 1 and 2 are just a couple of examples but the entire introduction of the manuscript needs to be re-written and carefully referenced to properly reflect the current state of the literature.

Response:

We thank the reviewer for the constructive suggestions. We have specified the plastic and elastic bending in the referenced examples, and re-written the whole introduction to better connect the mentioned examples with our system.

3) The authors of the present manuscript have made an attempt to determine the

mechanism of plastic flexibility in the current system using both micro focus SCXRD and Raman spectroscopy however in this case neither approach gives detailed and unequivocal structural confirmation of the mechanism because the XRD data could not be indexed and the Raman data alone does not provide enough information to do determine structures explicitly. This is OK – but the authors need to make it very clear that the proposed mechanism for plastic deformation is a hypothesis and not proven. In the concluding paragraph of the paper and in the abstract and in the title, the authors overstate the certainty of their findings.

Response:

We agree with the reviewer that we should made it clearer that our proposed mechanism is hypothesized based on our micro-focus XRD, Raman and SEM results. We have reorganized the title, abstract and conclusion in the revised manuscript.

4) The following sentences beginning on line 52 are very difficult to follow. “Specifically, the abundant former could allow molecules to rotate in a reversible manner under stress while the small number of latter are able to redistribute the applied mechanical strain *via* limited changes in the elastic regime. In the plastically bendable crystals, the irreversible deformation is mainly induced by the adaptive but unrecoverable changes of dispersive van der Waals and weakly directional C–H \cdots π interactions (ref 22).” Is there a clearer way to say all this? Note that the mechanism of plastic flexibility is almost always associated with dislocation along slip planes.

Response:

We have reorganized the corresponding text in the revised manuscript as the following:

“For the elastic bending, two general structural features are found in these crystals. The first is the existence of abundant isotropic weak interactions, such as dispersive van der Waals interactions and weak hydrogen bonds.^{12, 15, 32} The uniform and isotropic distribution of weak interactions in these crystals could provide an effective pathway to accommodate mechanical stress *via* small structural adaption in the elastic regime. In

addition, these elastically bendable crystals normally have an interlocked crystal packing motif, which discourages long range molecular sliding and lattice slippage, hence preventing plastic deformation and fracture over a significant range of stresses.^{20, 21,33}

For the plastic bending systems, the interaction distributions (the strong and weak interactions) are usually found to be anisotropic and directional in the crystal structures, facilitating the crystal sliding along the directions of the weak interactions and resulting in the final plastic deformation.^{4, 13, 25} These relatively weak interactions can be abundant in the lattice, often existing between adjacent layered molecular assemblies that are adhered *via* relatively strong directional bonds. Correspondingly, facile molecular sliding can be allowed in weak bonding directions while the structural scaffolding is maintained along the strong bonding orientation. Due to these interaction distributions, crystals can accommodate the imposed strain permanently through the slippage of the molecular layers past each other to maintain the crystal's integrity.^{4,13,34}

5) Line 90 – The authors indicate that the crystals remain bent after stress and that this “indicates plastic (or permanent) nature”. A relatively minor point but plastic and permanent are not interchangeable. Not all permanent deformations are plastic (e.g. if a crystal breaks or delaminates etc that process is not plastic but it is still permanent).

Response:

We have deleted “permanent” in the bracket to avoid any confusion.

6) Figure 1. (c) The picture of crystals bent in to letters to advertise a University has been done to death and adds no additional scientific value. What does bending the crystals into these letters show that bending them into any other letters or numbers or

any other random shape does not show?

Response:

We thank for the reviewer's suggestion and have removed the picture of crystals bent into the abbreviation of our university.

7) Line 113-114 shear modulus repeated.

Response: We have corrected this ignorance in the revised manuscript.

8) Line 125-6 The inability to index, solve and refine the X-ray data means all subsequent proposals for mechanisms are purely hypotheses. As stated earlier this needs to be fixed in the title, abstract, conclusions etc to make this clear throughout the MS.

Response:

We agree with the reviewer and have reorganized the title, abstract and conclusion to state the hypothesized mechanism in the revised manuscript.

9) There is some nice powder data presented on page 7. Can the authors comment on how much the shift in the peaks from a straight section of crystal to a bent section could be caused by delamination? Could delamination at the microscopic level result in peak broadening? What about the shift in peaks?

Response:

Considering the relatively large beam size ($\sim 70 \times 70 \mu\text{m}^2$) of our lab facility, the Bragg peak positioning and intensity integrating of any individual region (inner, middle or outer) in the bend is difficult to be precisely quantified. Therefore, we have removed the discussion about the 1D PXRD plots in the revised manuscript to avoid any confusion. Please see our detailed response to Question 2 raised by Reviewer 1. Nevertheless, we believe that the delamination leads to partial loss of long-range structural order and therefore could cause peak broadening.

10) Blue light is approx. 25000 cm^{-1} while red light is approx. 15000 cm^{-1} . With this in

mind almost all of the discussion about Raman spectra, peak positions and in particular peak shifts on page 9 of the MS is wrong. The concept of blue and red shifting can only make sense for spectral bands between 15000 and 25000 cm^{-1} . In the manuscript the peaks discussed are between 200 and 1300 cm^{-1} . For example, it makes no sense to say a shift from 1168 to 1173 cm^{-1} is “blue shifted” as such a peak is also shifting towards the red region of the spectrum and is therefore “red shifted” too. It would be more informative to talk about shifts to higher/lower energy or frequency.

Response:

We thank the reviewer for the constructive comment and have carefully revised our wording about frequency shifting of the Raman spectra in the revised manuscript.

11) The discussion of the results of the high-pressure crystallography perhaps needs some clarification. The authors note that the Cu-pi and pi-pi distances change markedly with pressure while the Cu-N bond remains virtually unchanged. I am not sure what the significance of this is in terms of plastic bending. This result is consistent with everything we know about crystallography. Contacts in which the interactions are described by broad and shallow potential energy wells e.g. pi-pi interactions are much more compressible than covalent and coordinate bonds that are described by deep narrow potential energy wells? Why is this relevant to the plastic deformation? There are no circumstances in which you would expect the Cu-N bond to significantly expand or contract enough to facilitate plastic flexibility.

Response:

Please see our response to Question 8 (minor concerns) raised by Reviewer 1 on page 12-13.

12) Sentence ending 198 – see other comments about being clear that the “adaptable reconstruction” is a hypothesis consistent with the data – not proven unequivocally.

Response:

We have made it clear that the “adaptable reconstruction” is hypothesized based on experimental data in the revised text.

13) Lines 201-204 I found this sentence confusing. Doesn't the process of delamination imply loss of crystal integrity? It is hard for me to see how delamination can take place while at the same time the crystal integrity (i.e. long range order) is retained?

Response:

We agree with the reviewer that the delamination implies the loss (at least part of loss) of crystal integrity. We have replaced "...preserve the crystal integrity..." with "...prevent the rupture of the crystal...".

14) Line 212-213 The authors state that bending requires elongation of the inner layers and shortening of the outer layers. I think this is around the wrong way? In any case this elongation and shortening only applies to elastic bending. In plastic crystals there is no requirement for the inside or outside arcs to lengthen or shorten.

Response: We agree with the reviewer that plastic bending does not require the crystal inside and outside to contract and elongate. We have deleted this sentence to avoid any confusion.

Supporting Information

Line 117 of the MS: The authors discuss results of micro-SCRD experiments in the MS but there are no details provided about the experimental set-up used in the supporting information. This information is vital. What was the cross-sectional diameter of the X-Ray beam? What was the relationship of the beam to the crystal? How many data sets collected and across what angle? The authors should provide sufficient detail so that the experiment could in principle be repeated by others. Also – without this information it is not possible to properly referee the results presented. For example, a highly focused beam say 10 microns orthogonal to the bent edge will give a very different result to a wider beam and or if it is oriented in an alternate direction with respect to the crystal.

Response:

The micro-SCXRD experimental details have been added in the Supporting

Information as the following:

Synchrotron measurements: “The synchrotron micro-focus X-ray diffraction measurements were conducted in the ID15B beam line ($\lambda = 0.41005 \text{ \AA}$) of the European Synchrotron Radiation Facility. The incident beam with a cross-section of $2 \times 2 \text{ \mu m}^2$ was normal to the bending plane of Cu-Trz crystals, and the diffraction images were collected at 10 equidistant positions with theta range of -10 to 10° from the inner, middle to the outer region of the bent with a 10 \mu m interval.”

Lab measurements: “The lab micro-focus X-ray diffraction was measured using a Rigaku XtaLAB MM007 CCD diffractometer (Cu $K\alpha$, $\lambda = 1.54056 \text{ \AA}$) with a focal spot size of $\sim 70 \times 70 \text{ \mu m}^2$. A carefully bent Cu-Trz crystal, with a length of $\sim 1 \text{ mm}$ and a width of $\sim 100 \text{ \mu m}$, was measured with an incident X-ray beam normal to the bending plane. The data collections were set with an exposure time of 0.5 s , in the theta angle range of 6 – 73° and the cell parameters were refined using the Olex2 software package. We have collected 1132 and 1356 diffraction frames for the straight and bent sections, respectively.”

As I said above this is nice work in a cutting-edge field. If the authors were to provide a new version addressing all the concerns above then I think the MS would be suitable for publication in Nature Comms.

Response:

We thank the reviewer again for the enthusiastic comments on the quality and novelty of our work.

Reviewer #3 (Remarks to the Author):

I have read this manuscript with interest. The paper contains much detailed and interesting data. It definitely deserves publication in a good journal. I would have no doubts to recommend it for publication in Crystal Growth and Design, CrystEngComm, IUCrJ, and many other top-journals. I do not see any reasons why this paper could not

be published also in Nature Communications: even though this is not the first example of a highly plastic crystal, that has been studied by a plethora of advanced experimental and computational techniques, such studies are still rare, very challenging, and always have a high impact.

My advise of improvement would be to add the discussion of a few references that seem to be quite relevant for this work:

Rath, B. B., & Vittal, J. J. (2021). Mechanical bending and modulation of photoactuation properties in a one-dimensional Pb (II) coordination polymer. *Chemistry of Materials*, 33(12), 4621-4627.

Thomas, S. P., Shi, M. W., Koutsantonis, G. A., Jayatilaka, D., Edwards, A. J., & Spackman, M. A. (2017). The Elusive Structural Origin of Plastic Bending in Dimethyl Sulfone Crystals with Quasi - isotropic Crystal Packing. *Angewandte Chemie International Edition*, 56(29), 8468-8472.

Pejov, L., Panda, M. K., Moriwaki, T., & Naumov, P. (2017). Probing structural perturbation in a bent molecular crystal with synchrotron infrared microspectroscopy and periodic density functional theory calculations. *Journal of the American Chemical Society*, 139(6), 2318-2328.

Arkhipov, S. G., Losev, E. A., Nguyen, T. T., Rychkov, D. A., & Boldyreva, E. V. (2019). A large anisotropic plasticity of L-leucinium hydrogen maleate preserved at cryogenic temperatures. *Acta Crystallographica Section B: Structural Science, Crystal Engineering and Materials*, 75(2), 143-151.

Feiler, T., Michalchuk, A. A., Schröder, V., List-Kratochvil, E., Emmerling, F., & Bhattacharya, B. (2021). Elastic Flexibility in an Optically Active Naphthalidenimine-Based Single Crystal. *Crystals*, 11(11), 1397.

Paliwoda, D., Wawrzyniak, P., & Katrusiak, A. (2014). Unwinding Au⁺··· Au⁺ bonded filaments in ligand-supported gold (I) polymer under pressure. *The Journal of Physical Chemistry Letters*, 5(13), 2182-2188.

Das, S., Saha, S., Sahu, M., Mondal, A., & Reddy, C. M. (2021). Temperature - Reliant

Dynamic Properties and Elasto - Plastic to Plastic Crystal (Rotator) Phase Transition in a Metal Oxyacid Salt. *Angewandte Chemie International Edition*, e202115359

Morritt, G. H., Michaels, H., & Freitag, M. (2022). Coordination polymers for emerging molecular devices. *Chemical Physics Reviews*, 3(1), 011306.

Response: We would like to thank the reviewer for these enthusiastic comments on the novelty and quality of our work. As the reviewer suggested, we have added the recommended references as Refs. 5, 12, 20, 21, 22, 24, 31, 35 and corresponding discussions on paragraph 1–2, page 3 in the introduction of the revised manuscript.

Ref. 12: Rath, B. B. & Vittal, J. J. Mechanical bending and modulation of photoactuation properties in a one-dimensional Pb(II) coordination polymer. *Chem. Mater.* **33**, 4621–4627 (2021).

Ref. 20: Thomas, S. P., et al. The elusive structural origin of plastic bending in dimethyl sulfone crystals with quasi - isotropic crystal packing. *Angew. Chem. Int. Ed.* **56**, 8468–8472 (2017).

Ref. 31: Pejov, L., Panda, M. K., Moriwaki, T., & Naumov, P. Probing structural perturbation in a bent molecular crystal with synchrotron infrared microspectroscopy and periodic density functional theory calculations. *J. Am. Chem. Soc.* **139**, 2318-2328 (2017).

Ref. 21: Arkhipov, S. G. et al. A large anisotropic plasticity of L-leucinium hydrogen maleate preserved at cryogenic temperatures. *Acta Crystallogr. Sect. B* **75**, 143-151 (2019).

Ref. 22: Feiler, T., et al. Elastic flexibility in an optically active naphthalidenimine-based single crystal. *Crystals* **11**, 1397 (2021).

Ref. 36: Paliwoda, D., Wawrzyniak, P., & Katrusiak, A. (2014). Unwinding Au⁺··· Au⁺ bonded filaments in ligand-supported gold (I) polymer under pressure. *J. Phys. Chem. Lett.* **5**, 2182-2188 (2014).

Ref. 24: Das, S. et al. Temperature - reliant dynamic properties and elasto - plastic to

plastic crystal (rotator) phase transition in a metal oxyacid salt. *Angew. Chem. Int. Ed.* **61**, e2021153 (2022).

Ref. 5: Morritt, G. H., Michaels, H., & Freitag, M. Coordination polymers for emerging molecular devices. *Chem. Phys. Rev.* **3**, 011306 (2022).

REVIEWER COMMENTS

Reviewer #1 (Remarks to the Author):

Thank you for the opportunity to re-review the manuscript by An et al on the mechanistic study of a plastically flexible coordination polymer single crystal. The authors have made significant efforts to address reviewer concerns, and have made really marked improvements in their manuscript. The work is certainly of interest to the community and will be well received. However, I note that significant advances have been already made in the field on this topic (many reported e.g. in Chem. Mater.), including evidence for delamination (i.e. the new mechanism proposed in the current paper), interdigitation, and detailed microfocus X-ray diffraction analysis. That said, the present paper is a very detailed and thorough investigation that will undoubtedly expand the thinking of the community. I would ask the authors to first consider a number of points, however:

The authors have significantly expanded the discussion of bending mechanisms in their introduction. However, their discussion focuses entirely on the concepts that were developed for molecular crystals, without any notable mention of the ongoing efforts on the study of coordination polymer flexibility. There have been already some major studies on these materials, for example by M Dakovic and B Bhattacharya. These studies have already discussed potential delamination mechanisms, potential interdigitation mechanisms, and have also employed both theoretical studies of inter-chain interaction energies and experimental microfocus X-ray diffraction studies to investigate bending. Note that the mechanisms and design criteria are not the same as for molecular solids. The authors should rework their paper in a way to better reflect the state of the art in understanding of the appropriate field.

I find it very curious that the bent region of the crystal never illuminates under polarized microscopy (Figure 1). Can the authors expand on this – is it in fact due to a loss of coherence in the structure, or simply due to the bent geometry of the crystal itself?

The SEM images from the authors are very interesting, and seem to suggest that any 'delamination' does not extend through the crystal, but rather is quite localized to the bent region. Does this suggest it is not really delamination, but rather buckling / interdigitation of the layers? Can the authors comment

The size of the beam used at ESRF seems to be inconsistent (2 x 2 or is it 2x4)?

The authors discuss high pressure diffraction results. However there is a recent paper by Liu et al (Nature Commun) that suggests that for plastically flexible CP crystals, hydrostatic compression is not a useful direction of investigation. Can the authors expand on the discussion of why they undertook these studies and its relevance to developing a mechanistic understanding of bending in CPs?

Reviewer #2 (Remarks to the Author):

The authors have responded to my queries from the original review appropriately. They also appear to have changed the paper in line with comments from other reviewers. If all reviewers are happy with the response from the authors then I recommend publishing this paper in Nature Communications.

REVIEWERS' COMMENTS

Reviewer #1 (Remarks to the Author):

Thank you for the opportunity to re-review the manuscript by An et al on the mechanistic study of a plastically flexible coordination polymer single crystal. The authors have made significant efforts to address reviewer concerns, and have made really marked improvements in their manuscript. The work is certainly of interest to the community and will be well received. However, I note that significant advances have been already made in the field on this topic (many reported e.g. in Chem. Mater.), including evidence for delamination (i.e. the new mechanism proposed in the current paper), interdigitation, and detailed microfocus X-ray diffraction analysis. That said, the present paper is a very detailed and thorough investigation that will undoubtedly expand the thinking of the community. I would ask the authors to first consider a number of points, however:

Response: We thank the reviewer's recognition and enthusiastic comments for our revised manuscript.

The authors have significantly expanded the discussion of bending mechanisms in their introduction. However, their discussion focuses entirely on the concepts that were developed for molecular crystals, without any notable mention of the ongoing efforts on the study of coordination polymer flexibility. There have been already some major studies on these materials, for example by M Dakovic and B Bhattacharya. These studies have already discussed potential delamination mechanisms, potential interdigitation mechanisms, and have also employed both theoretical studies of inter-chain interaction energies and experimental microfocus X-ray diffraction studies to investigate bending. Note that the mechanisms and design criteria are not the same as for molecular solids. The authors should rework their paper in a way to better reflect the state of the art in understanding of the appropriate field.

Response: We thank the reviewer for the constructive comments and pointing us to the works of Dakovic and Bhattacharya. We considered these papers and have included an additional paragraph in the introduction to better reflect the advances in the field of bendable coordination polymer crystals (page 4 in the revised manuscript).

“Although there are some common structural causes for the bending in molecular crystals and CPs, the bending mechanisms of CPs have been found to be distinct.^{4, 9-13} In particular, the low structural dimensionality of CPs often plays a pivotal role in achieving their bendability.^{4, 9} In most known systems, the low dimensional coordination chains act as rigid skeletons and are assembled via abundant weak molecular interactions along the other two directions. Upon bending, the coordination skeletons remain significantly less affected since the mechanical stress is mainly resolved via the reorganization of the weak bonding forces.”

I find it very curious that the bent region of the crystal never illuminates under polarized microscopy (Figure 1). Can the authors expand on this – is it in fact due to a loss of coherence in the structure, or simply due to the bent geometry of the crystal itself?

Response: We thank the reviewer for the valuable comments. Based on our synchrotron and laboratory micro-focus XRD, the diffraction from the bent region is greatly deteriorated, which

indicates that the bent region loses its single crystallinity. Due to the loss of single crystallinity, the bent region shows very trivial changes under the polarized light microscope. In other words, the disappearance of the systematic light extinction in the bent region is attributed to the loss of coherence in the structure.

The SEM images from the authors are very interesting, and seem to suggest that any ‘delamination’ does not extend through the crystal, but rather is quite localized to the bent region. Does this suggest it is not really delamination, but rather buckling / interdigitation of the layers? Can the authors comment

Response: The up-panel of **Fig. 1c** is cropped from the up-left SEM image in Supplementary Fig. 1c, so it is unable to show the whole view of the crystal bending. As seen from the updated up-panel image in Supplementary Fig. 1c (please see below), the delamination of the layers slightly extends to the straight regions which differs significantly from the layer buckling/interdigitation.

Up-left panel of Supplementary Fig. 1c | SEM images collected from a bent crystal. The yellow arrows indicate the extension of the delamination of the layers.

The size of the beam used at ESRF seems to be inconsistent (2 x 2 or is it 2x4)?

Response: The beam size is $2 \times 4 \mu\text{m}^2$ and we have corrected this ignorance in the revised manuscript.

The authors discuss high pressure diffraction results. However there is a recent paper by Liu et al (Nature Commun) that suggests that for plastically flexible CP crystals, hydrostatic compression is not a useful direction of investigation. Can the authors expand on the discussion of why they undertook these studies and its relevance to developing a mechanistic understanding of bending in CPs?

Response: We thank the reviewer for this constructive comments. The reported flexible crystal by Liu et al. is a tetragonal 1D coordination polymer (space group, $P-4b2$), in which adjacent coordination chains are linked via the same $\text{C-H}\cdots\text{Cl}$ bonds along the $[110]$ and $[-1-10]$ directions. Because of the quite isotropic packing mode in this crystal, it is not surprising that it shows a rather isotropic behavior under hydrostatic pressure. For Cu-Trz crystal studied in our work, it is monoclinic ($P2_1/c$) and has a very anisotropic packing mode. This special structural motif endows its highly anisotropic response to hydrostatic compression as shown in the Supplementary Figs 15&16. From the HP-XRD, the coordination chains constructed by Cu-N bonds show small changes (1.28%) while the $\text{Cu}\cdots\text{Cu}$ interactions and $\text{C-H}\cdots\pi$ interactions experience significant decrease (-

1.89% and -5.41%, respectively) upon compression. That is to say these weak interactions are amenable under mechanical stress and able to change adaptively to maintain the crystal's integrity.

Reviewer #2 (Remarks to the Author):

The authors have responded to my queries from the original review appropriately.

They also appear to have changed the paper in line with comments from other reviewers. If all reviewers are happy with the response from the authors then I recommend publishing this paper in Nature Communications.

Response: We are very pleased that all the concerns pointed by the reviewer have been well addressed. We thank the reviewer again for his/her valuable and constructive comments.